# Mouse SAS-6 is required for centriole formation in embryos and integrity in embryonic stem cells

**Marta Grzonka[1,2,3], Hisham Bazzi[1,2,4]\*†**

[1]Department of Cell Biology of the Skin and Department of Dermatology and Venereology, Medical Faculty, University of Cologne, Cologne, Germany; [2]The Cologne Cluster of Excellence in Cellular Stress Responses in Aging-associated Diseases (CECAD), Medical Faculty, University of Cologne, Cologne, Germany; [3]Graduate School for Biological Sciences, University of Cologne, Cologne, Germany; [4]Center for Molecular Medicine Cologne (CMMC), Medical Faculty, University of Cologne, Cologne, Germany

**\*For correspondence:**
hisham.bazzi@uk-koeln.de

**Present address:** †Cell & Developmental Biology, 3045 BSRB, University of Michigan Medical School, Ann Arbor, United States

**Competing interest:** The authors declare that no competing interests exist.

**Abstract** SAS-6 (SASS6) is essential for centriole formation in human cells and other organisms but its functions in the mouse are unclear. Here, we report that *Sass6*-mutant mouse embryos lack centrioles, activate the mitotic surveillance cell death pathway, and arrest at mid-gestation. In contrast, SAS-6 is not required for centriole formation in mouse embryonic stem cells (mESCs), but is essential to maintain centriole architecture. Of note, centrioles appeared after just one day of culture of *Sass6*-mutant blastocysts, from which mESCs are derived. Conversely, the number of cells with centrosomes is drastically decreased upon the exit from a mESC pluripotent state. At the mechanistic level, the activity of the master kinase in centriole formation, PLK4, associated with increased centriolar and centrosomal protein levels, endow mESCs with the robustness in using a SAS-6-independent centriole-biogenesis pathway. Collectively, our data suggest a differential requirement for mouse SAS-6 in centriole formation or integrity depending on PLK4 activity and centrosome composition.

## Editor's evaluation

This is an important study on the formation of mouse centrioles in the embryo and stem cells derived from them. The authors provide convincing evidence on the unique role of the SAS-6 protein and the entire cartwheel complex highlighting a mechanism that suggests a specific cell cycle related function of centriole formation and its maintenance.

## Introduction

Proliferating cells rely on stringent controls of cell division fidelity, primarily through the proper assembly, organization, and polarity of the microtubule-based mitotic spindle. In most animal cells, these functions are ensured by centrosomes, the major microtubule organizing centers (MTOCs). At the core of centrosomes are the microtubule-based centrioles that are highly conserved in evolution (*Bornens, 2012*; *Nabais et al., 2020*). During interphase or upon differentiation, centrioles provide the essential template to form cilia (*Conduit et al., 2015*). Cycling cells in G1 have two centrioles whose duplication is controlled by the centriole formation machinery, with the master kinase Polo-Like Kinase 4 (PLK4) regulating the early initiating steps (*Bettencourt-Dias et al., 2005*; *Habedanck et al., 2005*). Once per cycle, procentrioles assemble on the existing centrioles and form new daughter

centrioles (*Nigg and Holland, 2018*). The phenotypes of centriole loss vary between organisms and cell types depending on the essential function of centrioles in each context (*Marshall, 2007*). In humans, various mutations in genes encoding centrosomal and centriolar proteins cause primordial dwarfism and microcephaly (*Bond et al., 2005*; *Khan et al., 2014*). In the mouse, centrioles and centrosomes are crucial for embryonic development beyond mid-gestation (*Bazzi and Anderson, 2014a*; *Bazzi and Anderson, 2014b*). Null mutations in mouse *Cenpj* (also called *Sas-4*, *Sas4*, *Cpap*), which is essential for centriole formation, lead to the loss of centrioles and arrest by embryonic day (E) 9 (*Bazzi and Anderson, 2014a*). The acentriolar cells in the *Cenpj* null embryos activate a cell death pathway that is dependent on 53BP1 (gene name *Trp53bp1*), USP28 (gene name *Usp28*), and p53 (gene name *Trp53*), which is known as the mitotic surveillance pathway (*Lambrus and Holland, 2017*; *Xiao et al., 2021*).

Spindle assembly defective protein 6, SAS-6 (named SASS6), a core protein of the centriole biogenesis pathway (*Leidel et al., 2005*; *Nigg and Holland, 2018*), forms the major structural component of the cartwheel, which is the precursor ensuring a ninefold symmetry in centriole assembly and onto which microtubules of the forming procentriole are added (*Kitagawa et al., 2011*). SAS-6 consists of a conserved N-terminal head domain and an intrinsically disordered C-terminal region that flank a conserved coiled-coil domain whose homodimeric association forms a long tail. Hydrophobic interactions between the head domains of the dimers lead to the assembly of the hub, while the tails form the spokes of the cartwheel (*Hilbert et al., 2013*; *Kantsadi et al., 2022*; *Kitagawa et al., 2011*; *van Breugel et al., 2011*; *Yoshiba et al., 2019*). The loss of SAS-6 in the worm *C. elegans* and in human cell lines leads to centriole duplication failure and the consequent loss of centrioles, whereas in the algae *C. Reinhardtii* and flies *D. melanogaster*, mutations in *Sass6* result in centriole symmetry aberrations (*Leidel et al., 2005*; *Nakazawa et al., 2007*; *Rodrigues-Martins et al., 2007*).

Intriguingly, in early mouse development, centrioles form *de novo* around the blastocyst stage (E3.5) (*Courtois et al., 2012*; *Gueth-Hallonet et al., 1993*), from which mouse embryonic stem cells (mESCs) are derived and propagated. To our knowledge, there are currently no reports on the roles of SAS-6 in centriole assembly or integrity in mice or mouse cells. In this study, we asked whether SAS-6 is required for centriole *de novo* formation and duplication during mouse development and in mESCs. Our data show that the loss of SAS-6 in the developing mouse leads to centriole formation failure, activation of the 53BP1-, USP28-, and p53-dependent mitotic surveillance cell death pathway, and arrest of development at mid-gestation (E9.5). In contrast, mESCs without SAS-6 can still form centrioles, which are nonetheless structurally defective and lack the capacity to template cilia. While *Sass6* mutant blastocyst cells acquire centrioles in culture, mESCs exit from pluripotency leads to the loss of centrioles. Our data indicate that mESCs rely on PLK4 activity, and perhaps enriched centrosomal proteins, for their remarkable ability to bypass the requirement for SAS-6 in centriole duplication. Our findings highlight the importance of centrosome composition in centriole duplication, even for what are considered as core-duplication proteins like SAS-6.

**Table 1.** Description of CRISPR/Cas9-mediated knockouts of *Sass6* in the mouse *in vivo*.

| | *Sass6*[em4/em4] | *Sass6*[em5/em5] |
|---|---|---|
| Location | exon 4 | exon 5 |
| gRNA | 5'-GGTGGACTTCTTAGCTTTCC-3' | 5'-ACCGGTCCTTTTAAACGTAG-3' |
| Change | 3 bp del and 1 bp insertion (a net of 2 bp deletion) | 5 bp del |
| Mutation | NC_000069.7(Chr3)*: g.116399341_116399343delinsC | NC_000069.7(Chr3)*: g.116401034_116399338del |
| InDel | GGT<u>C</u>TTCTTAGCTTTCC | ACCGGTCCTTTTAAACG |
| Predicted STOP codon | 130 amino acids downstream of the translation start site, with 78 amino acids not native to the protein | 129 amino acids downstream of the translation start site, with six amino acids not native to the protein |

*RefSeq sequence number from GRCm39 assembly, NCBI annotation release 109.

## Results

### Mutation in mouse *Sass6* leads to embryonic arrest around mid-gestation

To determine the functions of mouse SAS-6 *in vivo*, we used CRISPR/Cas9 to generate *Sass6* knockout mice by targeting exon 4 (Materials and methods, *Table 1*). The resulting *Sass6* mutant allele (*Sass6^em4/em4*) had a frameshifting deletion, which is predicted to lead to a premature stop codon (*Table 1*). *Sass6^em4/em4* embryos arrested development ~E9.5, when they still formed a heart but did not show somites or undergo embryonic turning that are typical in wild-type (WT) embryos (*Figure 1A*). The phenotype of *Sass6^em4/em4* embryos resembled our previously reported *Cenpj^-/-* embryos without centrioles (*Bazzi and Anderson, 2014a*), suggesting a crucial role for SAS-6 in centriole formation and mouse development.

### The loss of SAS-6 activates the 53BP1-USP28-p53 mitotic surveillance pathway

In order to assess whether the loss of SAS-6 leads to p53 upregulation and cell death, as in *Cenpj^-/-* mutants (*Bazzi and Anderson, 2014a*), we performed immunostaining for p53 and active cleaved-caspase 3 (Cl. CASP3) on sections from WT and *Sass6^em4/em4* embryos at E8.5 (*Figure 1B and C*). In comparison to WT embryos where both were rarely detectable, the *Sass6^em4/em4* mutants showed significantly increased levels of nuclear p53 (~2.5 fold) and Cl. CASP3 (~2.5 fold) (*Figure 1B–E*). To determine whether the stabilization of p53 and increased cell death in *Sass6^em4/em4* embryos were associated with prolonged mitoses as in *Cenpj^-/-* mutants (*Bazzi and Anderson, 2014a*), sections from E8.5 WT and *Sass6^em4/em4* embryos were immune-stained for the mitotic marker phospho-histone H3. In agreement, *Sass6^em4/em4* embryos showed a significant increase in the fraction of mitotic cells (10%) compared to WT (5%) (*Figure 1—figure supplement 1*).

Next, we asked whether the *Sass6* mutant phenotype is caused by the activation of the p53-, 53BP1-, and USP28-dependent mitotic surveillance pathway (*Xiao et al., 2021*). To functionally address this question, we crossed *Sass6^+/em4* mice to *Trp53^+/-*, *Trp53bp1^+/-*, or *Usp28^+/-* null mouse alleles (*Marino et al., 2000*; *Xiao et al., 2021*). All three double-mutant embryos: *Sass6^em4/em4 Trp53^-/-*, *Sass6^em4/em4 Trp53bp1^-/-*, and *Sass6^em4/em4 Usp28^-/-*, were evidently rescued at E9.5 as judged by the normalized size and morphology, which were more similar to WT than to the *Sass6^em4/em4* single-mutant embryos (*Figure 1A*). In this regard, the double-mutant embryos showed body turning and visible somites, and were also similar to our reported mitotic surveillance pathway double mutants with *Cenpj* (*Xiao et al., 2021*). In line with the rescue of embryo morphology, double-mutant embryos also showed significantly reduced levels of p53 and Cl. CASP3 (*Figure 1B–E*). The data indicated that, similar to SAS-4, the loss of SAS-6 in the mouse activates the mitotic surveillance pathway leading to cell death and embryonic arrest at mid-gestation.

### Mouse SAS-6 is essential for centriole formation *in vivo*

To characterize the *Sass6^em4/em4* mutant embryos for *Sass6* expression (*Figure 1A*, *Figure 2A*), we performed Western blotting using a SAS-6-specific antibody on WT and *Sass6^em4/em4* embryo lysates at E9 (*Figure 2B*). The levels of SAS-6 in *Sass6^em4/em4* embryos were drastically reduced compared to WT (*Figure 2B*). Using the same SAS-6 antibody combined with the centrosomal marker γ tubulin (TUBG), we also performed immunofluorescence analyses on WT and *Sass6^em4/em4* embryo sections at E9 (*Figure 2C*). SAS-6 co-localized with TUBG in almost all cells of WT (95%) and in 19% of cells in *Sass6^em4/em4* (*Figure 2C and D*). These findings indicated that the residual SAS-6 in *Sass6^em4/em4* embryos was due to the allele being hypomorphic for *Sass6*. Thus, we generated another *Sass6* mutant allele with a frameshifting deletion in exon 5 (*Sass6^em5/em5*), which is predicted in silico to result in a premature stop codon (*Table 1*). *Sass6^em5/em5* embryos arrested development at E9.5 and morphologically resembled *Sass6^em4/em4* mutants (*Figure 2A*). Western blot showed that SAS-6 is not detectable in *Sass6^em5/em5* mutant embryos compared to WTs (*Figure 2B*). In addition, immunostaining for SAS-6 and TUBG showed that SAS-6 co-localized with TUBG only rarely in *Sass6^em5/em5* (4%) (*Figure 2C and D*), suggesting that it was a stronger loss-of-function allele than *Sass6^em4/em4*.

To examine whether SAS-6 is required for centriole formation in the mouse, we crossed the *Sass6^+/em4* or *Sass6^+/em5* alleles to Cetn2-eGFP, a transgenic mouse line where centrioles are marked with

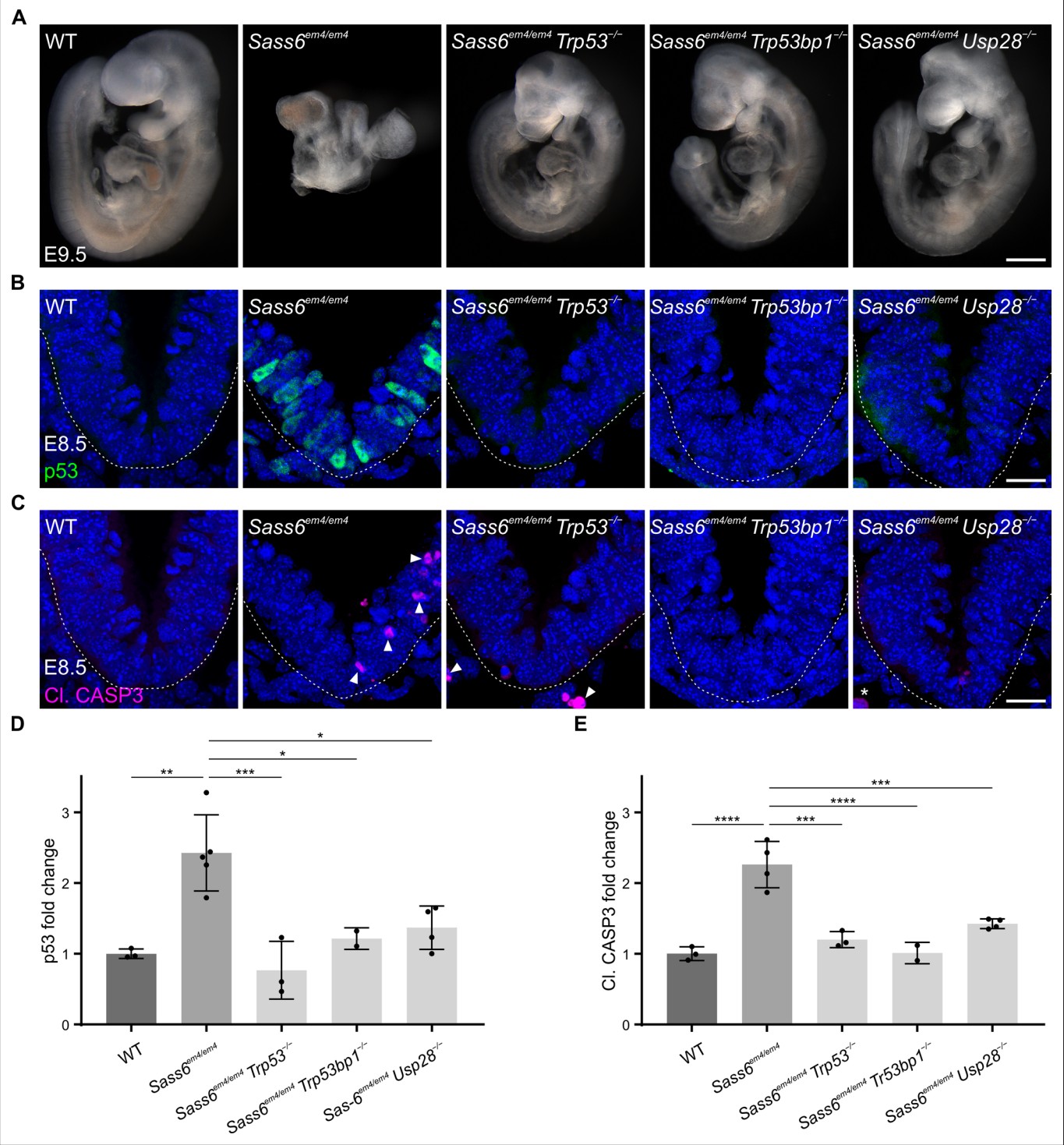

**Figure 1.** Mutation in mouse *Sass6* activates the 53BP1-USP28-p53 mitotic surveillance pathway. (**A**) Left-side views of wild-type (WT), *Sass6*$^{em4/em4}$, *Sass6*$^{em4/em4}$ *Trp53*$^{-/-}$, *Sass6*$^{em4/em4}$ *Trp53bp1*$^{-/-}$, and *Sass6*$^{em4/em4}$ *Usp28*$^{-/-}$ embryos at E9.5. Anterior is up in all images. At least three embryos per genotype showed similar phenotypes. Scale bar = 500 µm. (**B**) Immunostaining for p53 on transverse sections of WT, *Sass6*$^{em4/em4}$, *Sass6*$^{em4/em4}$ *Trp53*$^{-/-}$, *Sass6*$^{em4/em4}$ *Trp53bp1*$^{-/-}$, and *Sass6*$^{em4/em4}$ *Usp28*$^{-/-}$ embryos at E8.5. The sections shown encompass the neural plate (top) and mesenchyme (bottom), demarcated by the dashed line. Dorsal is up in all images. Scale bars = 25 µm. (**C**) Immunostaining for Cleaved-Caspase3 (Cl. CASP3) as mentioned in (**B**). Arrowheads indicate Cl. CASP3-positive cells, while asterisks mark non-specific staining of blood cells. (**D**) Quantification of the nuclear p53 in (**B**). Values were normalized to WT. Error bars represent mean ± SD WT: 1.00 ± 0.06 (n=2582 cells from three embryos); *Sass6*$^{em4/em4}$: 2.4 ± 0.5 (n=2372 from four embryos); *Sass6*$^{em4/em4}$ *Trp53*$^{-/-}$: 0.8 ± 0.3 (n=2379 from three embryos); *Sass6*$^{em4/em4}$ *Usp28*$^{-/-}$: 1.4 ± 0.3 (n=2775 from four embryos);

*Figure 1 continued on next page*

Figure 1 continued

*Sass6*<sup>em4/em4</sup> *Trp53bp1*<sup>−/−</sup>: 1.2 ± 0.1 (n=1840 from two embryos). *p<0.05, **p<0.01, ***p<0.001 (one-way ANOVA with Tukey's multiple comparisons). (**E**) Quantification of Cl. CASP3 in (**C**) as mentioned in (**D**). WT: 1.00 ± 0.1 (n=2582 cells from three embryos); *Sass6*<sup>em4/em4</sup>: 2.3 ± 0.3 (n=2372 from four embryos); *Sass6*<sup>em4/em4</sup> *Trp53*<sup>−/−</sup>: 1.2 ± 0.1 (n=2379 from three embryos); *Sass6*<sup>em4/em4</sup> *Usp28*<sup>−/−</sup>: 1.4 ± 0.1 (n=2775 from four embryos); *Sass6*<sup>em4/em4</sup> *Trp53bp1*<sup>−/−</sup>: 1 ± 0.1 (n=1840 from two embryos). ***p<0.001, ****p<0.0001 (one-way ANOVA with Tukey's multiple comparisons).

The online version of this article includes the following figure supplement(s) for figure 1:

**Figure supplement 1.** Mutations in *Sass6* lead to an increase in the mitotic index in mouse embryos.

centrin-eGFP (*Bangs et al., 2015*; *Higginbotham et al., 2004*), and stained embryo sections at E9.0 for TUBG (*Figure 2E*). Centrosomes positive for both CETN2-eGFP and TUBG were present in almost all of the cells in control Cetn2-eGFP embryos (98%) (*Figure 2E and F*). In contrast, centrioles were identified only in a minor fraction of cells in *Sass6*<sup>em4/em4</sup> Cetn2-eGFP mutants (6%) and even less in *Sass6*<sup>em5/em5</sup> Cetn2-eGFP (2%) (*Figure 2E and F*). In addition, we immunostained *Sass6*<sup>em4/em4</sup> and *Sass6*<sup>em5/em5</sup> embryo sections at E9.0 with CEP164, a mother centriole distal appendage marker, and TUBG (*Figure 2—figure supplement 1A*). We detected centrioles, as defined by the co-localization of TUBG and CEP164, in almost all of the cells in WT embryos (97%), in a small fraction of cells in *Sass6*<sup>em4/em4</sup> mutants (16%) but not in *Sass6*<sup>em5/em5</sup> (*Figure 2—figure supplement 1A, B*).

For higher resolution analyses of centriole formation in *Sass6* mutant embryos, we utilized Ultrastructure-Expansion Microscopy (U-ExM), a technique that relies on isometrically expanding the sample ~4 times and has been recently widely implemented for centriole analyses (*Gambarotto et al., 2019*). We combined U-ExM with immunostaining for the centriolar wall marker, acetylated tubulin (Ac-TUB) (*Figure 2G*). We observed that in WT embryo sections (E9.0), each pole of a mitotic spindle contained a pair of centrioles, while in *Sass6*<sup>em4/em4</sup> mutants, only rare and single centrioles were detected (11%), suggesting centriole duplication failure (*Figure 2G and H*). Of note, no centrioles were detected in the mitotic poles of *Sass6*<sup>em5/em5</sup> embryos (*Figure 2G and H*). Overall, the data suggested that the *Sass6*<sup>em4/em4</sup> is a severe hypomorphic allele of *Sass6*, whereas the *Sass6*<sup>em5/em5</sup> is likely to be a null allele of *Sass6*, and that SAS-6 is essential for centriole formation in mouse embryos *in vivo*.

## SAS-6 is required for centriole integrity, but not formation, in mESCs

To study the roles of *Sass6* in an *in vitro* setting that mimics mouse embryonic development, we chose to knockout *Sass6* in mESCs. To accomplish this without any reasonable doubt of residual SAS-6 protein, we used CRISPR/Cas9 with a pair of guide RNAs (gRNAs) flanking the open reading frame (ORF) of *Sass6*, and engineered a null allele lacking the entire *Sass6* ORF (*Sass6*<sup>−/−</sup>) (*Figure 3A*, Materials and methods, *Table 2*). The deletion of the *Sass6* ORF in *Sass6*<sup>−/−</sup> mESCs was confirmed at the level of DNA using PCR (*Figure 3A*, bottom panel), the loss of *Sass6* mRNA validated by RT-PCR (*Figure 3B*), and the lack of detectable SAS-6 protein corroborated by Western blot (*Figure 3C*) and immunostaining (*Figure 3—figure supplement 1A*). We next used immunostaining for TUBG and FOP, a centriole marker, to assess centrosome and centriole formation in *Sass6*<sup>−/−</sup> mESCs. While centrosomes, as defined by the co-localization of TUBG and FOP, were evident in the vast majority of WT mESCs (94%), it was surprising that more than half of *Sass6*<sup>−/−</sup> mESCs still possessed centrosomes (56%) (*Figure 3D and E*). This unexpected observation suggested that unlike in mouse embryos, SAS-6 may not be essential for centrosome formation in mESCs.

To probe whether the TUBG-marked centrosomes in *Sass6*<sup>−/−</sup> mESCs contained centrioles at their core, we used higher-resolution U-ExM combined with immunostaining for *bona fide* centriolar wall markers (Ac-Tub and α/β-tubulin, TUB) (*Figure 3F*). Almost all of the centrosomes in WT cells contained two or more centrioles (99%) and only a rare fraction contained one centriole (1%) (*Figure 3F and G*). In contrast, in *Sass6*<sup>−/−</sup> mESCs, about half of the centrosomes had two or more centrioles (46%) and a comparable fraction had one centriole (45%) (*Figure 3F and G*). Additionally, in a minor fraction of *Sass6*<sup>−/−</sup> centrosomes (9%), aberrant centriolar threads were detected (*Figure 3F and G*). Structurally, we classified the centrioles in *Sass6*<sup>−/−</sup> mESCs into the following three categories: normal-like centrioles (18%), abnormal centrioles (65%), and thread-like structures (17%) (*Figure 3F and H*). Quantitatively, we measured the length of the normal-like centrioles and found that they were significantly longer in in *Sass6*<sup>−/−</sup> than in WT (*Figure 3I*). In summary, the data suggested that *Sass6*<sup>−/−</sup> centrioles in mESCs had a compromised ability to duplicate and/or were unstable with mostly abnormal structures.

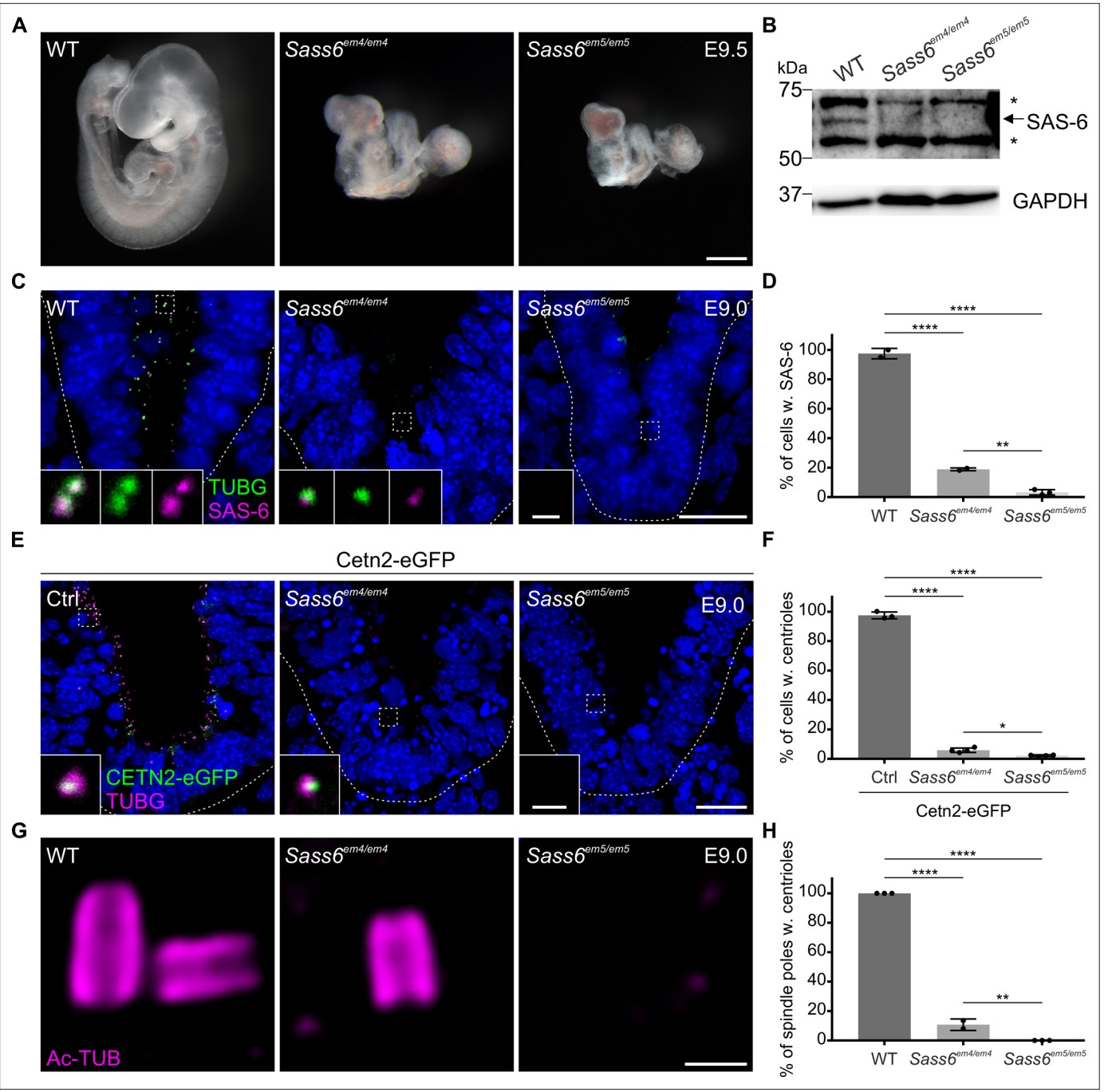

**Figure 2.** *Sass6^{em4/em4}* are severe hypomorphs while *Sass6^{em5/em5}* embryos lack centrioles. (**A**) Left-side views of wild-type (WT), *Sass6^{em4/em4}*, and *Sass6^{em5/em5}* embryos at E9.5. Anterior is up in all images. At least five embryos were analyzed per genotype. Scale bar = 500 μm. (**B**) Western blot analysis using a SAS-6-specific antibody on E9.5 WT, *Sass6^{em4/em4}*, and *Sass6^{em5/em5}* embryo extracts. Asterisks mark non-specific bands. GAPDH is used as a loading control. (**C**) Immunostaining for TUBG and SAS-6 on sagittal sections of WT, *Sass6^{em4/em4}*, and *Sass6^{em5/em5}* embryos at E9.0. The sections shown encompass the neural plate (top) and mesenchyme (bottom), demarcated by the dashed line. Insets are magnifications of the center of the dashed squares. Dorsal is up in all images. Scale bars = 20 μm and 1 μm (insets). (**D**) Quantification of the percentage of cells with SAS-6 signal co-localization with TUBG in (**C**). Error bars represent mean ± SD WT: 95 ± 3% (n=1929 cells from three embryos); *Sass6^{em4/em4}*: 19 ± 1% (n=542 from two embryos); *Sass6^{em5/em5}*: 4 ± 2% (n=2458 from four embryos). ****p<0.0001, **p<0.01 (one-way ANOVA with Tukey's multiple comparisons). (**E**) Immunostaining for TUBG on transverse sections of Cetn2-eGFP, *Sass6^{em4/em4}* Cetn2-eGFP, and *Sass6^{em5/em5}* Cetn2-eGFP embryos at E9.0. The sections shown are similar to those described in (**C**). Insets are magnifications of the center of the dashed squares. Scale bars = 20 μm and 1 μm (insets). (**F**) Quantification of the percentage of cells with centrioles (TUBG and Centrin-eGFP) is shown in (**E**). Error bars represent mean ± SD Cetn2-eGFP: 98 ± 2% (n=11,196 cells from three embryos); *Sass6^{em4/em4}* Cetn2-eGFP: 6 ± 1% (n=9752 from four embryos); *Sass6^{em5/em5}* Cetn2-eGFP: 2 ± 0.5% (n=5559 from four embryos).

*Figure 2 continued on next page*

*Figure 2 continued*

****p<0.0001, *p<0.05 (one-way ANOVA with Tukey's multiple comparisons). (**G**) Immunostaining for Ac-TUB on U-ExM sections from E9.0 embryos of the indicated genotypes. Scale bar = 200 nm. (**H**) Quantification of the percentage of mitotic spindle poles with centrioles in (**G**). Error bars represent mean ± SD WT: 100 ± 0% (n=65 spindle poles from three embryos); *Sass6*^em4/em4^: 11 ± 0.03% (n=62 from two embryos); *Sass6*^em5/em5^: 0 ± 0% (n=45 from three embryos). ****p<0.0001, **p<0.01 (one-way ANOVA with Tukey's multiple comparisons).

The online version of this article includes the following source data and figure supplement(s) for figure 2:

**Source data 1.** Western blot analysis on embryos.

**Figure supplement 1.** *Sass6*^em5/em5^ embryos lack mother centrioles marked with CEP164.

We next asked whether mESCs derived from *Sass6*^em5/em5^ mutant blastocysts can also form centrioles like *Sass6*^−/−^ mESCs. Therefore, we derived and propagated mESCs from *Sass6*^+/em5^ Cetn2-eGFP (controls) and *Sass6*^em5/em5^ Cetn2-eGFP (mutant) blastocysts at E3.5 and immunostained for TUBG (*Figure 3—figure supplement 1B*). Similar to *Sass6*^−/−^ mESCs, three-quarters of *Sass6*^em5/em5^ Cetn2-eGFP mESCs had centrosomes as defined by co-localization of CETN2-eGFP and TUBG (76%) (*Figure 3—figure supplement 1B, C*). Numerically, we used U-ExM combined with Ac-TUB and GFP immunostaining and showed that around half of the *Sass6*^em5/em5^ Cetn2-eGFP centrosomes had two or more centrioles (53%) and one-third had only one centriole (33%) (*Figure 3—figure supplement 1D, E*). In addition, a smaller fraction of centrosomes in *Sass6*^em5/em5^ Cetn2-eGFP mESCs showed centriolar threads (14%) (*Figure 3—figure supplement 1D, E*). At the structural level, almost half of *Sass6*^em5/em5^ Cetn2-eGFP centrioles were abnormal (48%), about a third exhibited normal-like centrioles (31%) and the rest exhibited thread-like structures (21%) (*Figure 3—figure supplement 1D, F*). The data on centriole number and structure from *Sass6*^em5/em5^ Cetn2-eGFP mESCs was largely in agreement with that of *Sass6*^−/−^ mESCs, suggesting that regardless of the method or timing of SAS-6 removal, mESCs retained the capacity to form centrioles independently of SAS-6.

## The centrioles in *Sass6*^−/−^ mESCs have proximal and distal defects

SAS-6 has been shown to cooperate with STIL, an essential component for centriole duplication, to initiate procentriole formation (*Kratz et al., 2015*). To address whether the fraction of centrioles that failed to duplicate in *Sass6*^−/−^ mESCs is associated with an impairment of STIL recruitment, we used U-ExM combined with Ac-TUB and STIL immunostaining (*Figure 4A*, *Figure 4—figure supplement 1A*). STIL localized to three-quarters of centrosomes in WT mESCs undergoing centriole duplication (74%) (*Figure 4A and B*; *Figure 4—figure supplement 1A*). In *Sass6*^−/−^ mESCs, STIL localized to less than one-third of the centrosomes (29%) (*Figure 4A and B*; *Figure 4—figure supplement 1A*), which is roughly half of the STIL-positive centrosomes in WT cells, and might account for the duplication failure in almost half of the *Sass6*^−/−^ centrosomes (single centrioles in *Figure 3G*). To assess whether the loss of centriole integrity in *Sass6*^−/−^ mESCs is associated with disruption of the proximal centriole end or internal structural scaffold, we immunostained centrioles in U-ExM for CEP135 (proximal end) and POC5 (scaffold) (*Figure 4C and E*; *Figure 4—figure supplement 1B*). We found that CEP135 localized to the proximal centriole in all WT cells, while the number of centrioles with CEP135 was decreased in *Sass6*^−/−^ cells (73%) (*Figure 4C and D*). Notably, only a minority of CEP135-positive centrioles in *Sass6*^−/−^ mESCs showed normal CEP135 localization (12%) (*Figure 4—figure supplement 1B, C*). On the other hand, POC5 was present in WT and *Sass6*^−/−^ intact centrioles, but also lining the abnormal centrioles and centriolar threads in *Sass6*^−/−^ cells, further confirming their centriolar nature (*Figure 4E*).

To investigate whether centrioles in *Sass6*^−/−^ mESCs exhibit distal-end capping defects, we combined U-ExM with immunostaining for the distal cap proteins, CP110 and CEP97 (*Figure 4F*; *Figure 4—figure supplement 1D*). The majority of both WT (91%) and *Sass6*^−/−^ (82%) centrosomes had CP110 present on the centrioles' distal end (*Figure 4F and G*); However, the fraction of centrosomes with centrioles associated with CEP97 was slightly decreased in *Sass6*^−/−^ mESCs (73%) compared to WT (95%) (*Figure 4—figure supplement 1D, E*).

Next, we examined whether the mother centrioles in *Sass6*^−/−^ mESCs were decorated with distal appendages and used U-ExM combined with Ac-TUB and CEP164 immunostaining (*Figure 4H*; *Figure 4—figure supplement 1F*). The data showed, as expected, that CEP164 mostly localized to the mother centrioles in WT centrosomes (94%) (*Figure 4H,I*; *Figure 4—figure supplement 1F*). In

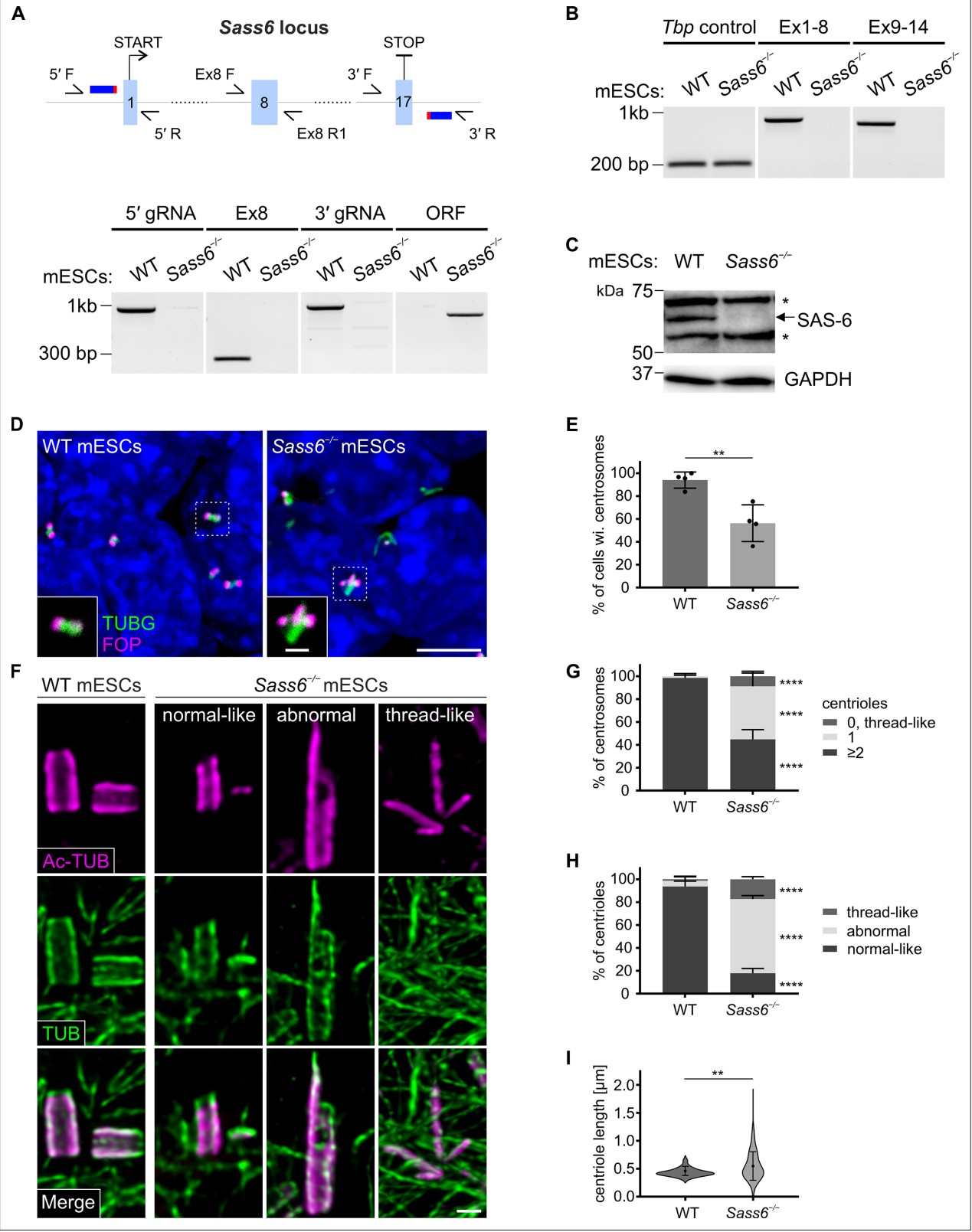

**Figure 3.** SAS 6 is required for centriole integrity, but not formation, in mouse embryonic stem cells (mESCs). (**A**) (Top) Schematic showing the CRISPR/Cas9 strategy using two gRNAs to delete the entire *Sass6* open reading frame (ORF) in mESCs. Exons (Ex) are represented by light blue boxes, gRNAs by dark blue thick horizontal lines, and PAM sites in red. Half arrows indicate the primers used for PCR analyses (below). (Bottom) Confirmation of the *Sass6* deletion in *Sass6⁻/⁻* mESCs by genomic PCR. The picture shows the PCR products using the following primers indicated in the schematic above: 5′

*Figure 3 continued on next page*

*Figure 3 continued*

gRNA (5' F and 5' R, band = 977 bp), Ex8 (Ex8 F and Ex8 R1, band = 281 bp), 3' gRNA (3' F and 3' R, band = 992 bp), *Sass6* ORF (5' F and 3' R, 825 bp in *Sass6*⁻/⁻, 34,349 bp in wild-type (WT), product too long to be amplified). (**B**) RT-PCR analyses of *Sass6* transcripts in WT and *Sass6*⁻/⁻ mESCs. The picture shows the PCR products from RT-PCR using the following primers: from Ex1 to Ex8 (Ex1 F and Ex8 R2, band = 734 bp), from Ex9 to Ex14 (Ex9 F and Ex14 R, band = 617 bp), *Tbp* Ctrl (Tbp F and Tbp R, band = 156 bp). (**C**) Western blot analysis using a SAS-6-specific antibody on WT and *Sass6*⁻/⁻ mESCs extracts. Asterisks mark non-specific bands. GAPDH is used as a loading control. (**D**) Immunostaining for TUBG and FOP in WT and *Sass6*⁻/⁻ mESCs. Insets are magnifications of the center of the dashed squares. Scale bars = 5 μm and 1 μm (insets). (**E**) Quantification of the percentage of cells with centrosomes (TUBG and FOP) in (**D**) from four independent experiments. Error bars represent mean ± SD WT: 94 ± 6% (n=2450 cells); *Sass6*⁻/⁻: 56 ± 14% (n=2766). **p<0.01 (two-tailed Student's t-test). (**F**) Centrioles were visualized using U-ExM and immunostaining for α- and β-tubulin (TUB) and Ac-TUB in WT and *Sass6*⁻/⁻ mESCs. Scale bar = 200 nm. (**G**) Quantification of the percentage of centrosomes with ≥2, 1, or 0 centrioles in (**F**) in WT and *Sass6*⁻/⁻ mESCs from five independent experiments. Error bars represent mean ± SD WT (n=156 centrosomes):≥2 centrioles = 99 ± 2%; 1 centriole = 1 ± 2%; *Sass6*⁻/⁻ (n=254):≥2 centrioles = 45 ± 8%, 1 centriole = 46 ± 10%, 0 centrioles = 9 ± 4%. ****p<0.0001 (two-tailed Student's t-test) (**H**) Quantifications of the percentage of centrioles within each category in (**F**) from five independent experiments. Error bars represent mean ± s.d. WT (n=330 centrioles): normal-like centrioles = 94 ± 4%; abnormal centrioles = 5 ± 3%; thread-like structures = 1 ± 2%; *Sass6*⁻/⁻ (n=432): normal-like centrioles = 18 ± 4%, abnormal centrioles = 65 ± 3%, thread-like structures = 17 ± 2%. ****p<0.0001, (two-tailed Student's t-test). (**I**) Violin plots of centriole length of normal-like centrioles in (**F**) in WT and *Sass6*⁻/⁻ mESCs from five independent experiments. Error bars represent mean ± SD WT: 0.46 ± 0.07 μm (n=72 centrioles); *Sass6*⁻/⁻: 0.55 ± 0.25 μm (n=72). **p<0.01 (two-tailed Student's t-test).

The online version of this article includes the following source data and figure supplement(s) for figure 3:

**Source data 1.** PCR, RT-PCR, and Western blot analyses on mouse embryonic stem cells (mESCs).

**Figure supplement 1.** SAS-6 is not essential for centriole formation in mouse embryonic stem cells (mESCs).

contrast, only a quarter of the centrosomes in *Sass6*⁻/⁻ mESCs had centrioles associated with CEP164 (28%) (***Figure 4H,I***; ***Figure 4—figure supplement 1F***). To assess whether the abnormal centrioles in *Sass6*⁻/⁻ mESCs retained the ability to template cilia, we used immunostaining against the ciliary axoneme marker Ac-TUB and ciliary membrane protein ARL13B (***Figure 4J***). Although cilia were present in only a small fraction of WT mESCs (11%), no cilia were detected in *Sass6*⁻/⁻ mESCs (***Figure 4J and K***), suggesting that SAS-6 is not only required for centriole integrity, but also distal appendage recruitment and cilia formation in mESCs.

## Short-term culture of *Sass6*^em5/em5^ blastocysts induces centriole formation

The finding that *Sass6*^em5/em5^ Cetn2-eGFP mESCs derived from E3.5 blastocysts are also able to form centrioles, prompted us to investigate whether these centrioles formed *de novo* in the absence of SAS-6. To begin to address this question, we combined the Cetn-eGFP with TUBG immunostaining in control Cetn2-eGFP blastocysts, which showed that the majority of cells had foci positive for both markers (73%) (***Figure 5A and B***). In contrast, in mutant *Sass6*^em5/em5^ Cetn2-eGFP blastocysts, only small TUBG accumulations were observed in a quarter of the cells (23%), but they did not contain CETN2-eGFP, suggesting that they were devoid of centrioles (***Figure 5A and B***). We next asked how early the centrioles form during the derivation of mESCs from the *Sass6*^em5/em5^ Cetn2-eGFP blastocysts. After 24 hr (hr) of culture, almost all of the cells in the cultured WT blastocysts contained centrioles (98% with both CETN2-eGFP and TUBG), and remarkably, centrioles were already detectable in one-third

**Table 2.** Description of CRISPR/Cas9 mediated knockout of *Sass6* in mouse embryonic stem cells (mESCs).

| | *Sass6*⁻/⁻ mESCs |
| --- | --- |
| Location | Deletion of entire ORF of *Sass6* |
| gRNAs | 5'-TAACAAACGTGGCCGCCTGA-3' <br> 5'-ACCAAGCCTGAGTTACACAA-3' |
| Change | 34,524 bp deletion |
| Mutation | NC_000069.7(Chr3)*:g.116388519_116423042del |
| Predicted protein | No predicted protein expression |

*RefSeq sequence number from GRCm39 assembly, NCBI annotation release 109.

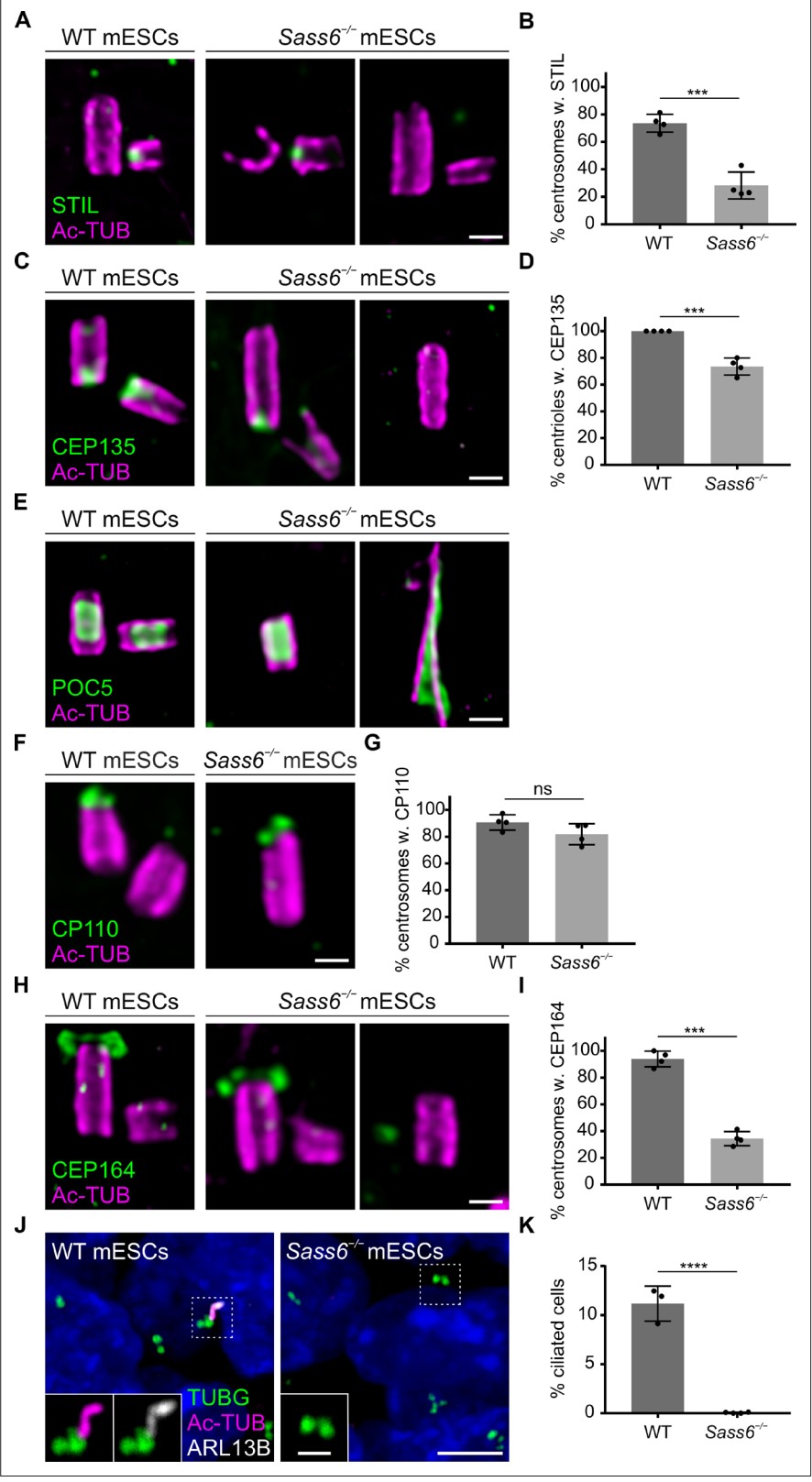

**Figure 4.** Centrioles in *Sass6*−/− mouse embryonic stem cells (mESCs) exhibit proximal and distal defects.
(**A**) Immunostaining for Ac-TUB and STIL of U-ExM of centrioles from wild-type (WT) and *Sass6*−/− mESCs.
Examples of centrioles with or without STIL are shown. Scale bar = 200 nm. (**B**) Quantification of the percentage of
centrosomes with (w.) STIL in (**A**) from four independent experiments. Error bars represent mean ± SD WT: 74 ± 6%

*Figure 4 continued on next page*

*Figure 4 continued*

(n=72 centrosomes); *Sass6⁻/⁻*: 29 ± 8% (n=94). ***p<0.001, (two-tailed Student's t-test). (**C**) Immunostaining for Ac-TUB and cartwheel protein (CEP135) of U-ExM of centrioles from WT and *Sass6⁻/⁻* mESCs. Examples of centrioles with or without CEP135 are shown. Scale bar = 200 nm. (**D**) Quantifications of the percentage of centrioles with CEP135 in (**C**) from four independent experiments. Error bars represent mean ± SD WT: 100 ± 0% (n=160 centrioles); *Sass6⁻/⁻*: 73 ± 6% (n=98). ***p<0.001, (two-tailed Student's t-test). (**E**) Immunostaining for Ac-TUB and the inner scaffold protein POC5 of U-ExM of centrioles from WT and *Sass6⁻/⁻* mESCs. Examples of normal-like or abnormal centrioles with POC5 are shown. Scale bar = 200 nm. (**F**) Immunostaining for Ac-TUB and the distal-end capping protein CP110 of U-ExM of centrioles from WT and *Sass6⁻/⁻* mESCs. Scale bar = 200 nm. (**G**) Quantification of the percentage of centrosomes with CP110 in (**F**) from four independent experiments. Error bars represent mean ± SD WT: 91 ± 5% (n=116 centrosomes); *Sass6⁻/⁻*: 82 ± 7% (n=106). ns = not significant with p>0.05 (two-tailed Student's t-test). (**H**) Immunostaining for Ac-TUB and CEP164 of U-ExM of centrioles from WT and *Sass6⁻/⁻* mESCs. Examples of centrioles with or without CEP164 are shown. Scale bar = 200 nm. (**I**) Quantification of the percentage of centrosomes with mother centrioles (Ac-TUB) with the distal appendage marker (CEP164) in (**H**). Error bars represent mean ± SD WT: 94 ± 5% (n=104 centrosomes from four independent experiments); *Sass6⁻/⁻*: 28 ± 14% (n=140 from five experiments). ***p<0.001 (two-tailed Student's t-test). (**J**) Immunostaining of the cilia markers ARL13B and Ac-TUB, and basal bodies marked with TUBG, on WT and *Sass6⁻/⁻* mESCs. The insets show separate channels for the magnifications of the center of the dashed squares. Scale bars = 5 μm and 1 μm (insets). (**K**) Quantification of the percentage of ciliated cells in (**J**). Error bars represent mean ± SD WT: 11 ± 1% (n=2602 cells from three experiments); *Sass6⁻/⁻*: 0 ± 0% (n=4602 from four experiments). ****p<0.0001 (two-tailed Student's t-test).

The online version of this article includes the following figure supplement(s) for figure 4:

**Figure supplement 1.** Centrioles in *Sass6⁻/⁻* mouse embryonic stem cells (mESCs) exhibit structural abnormalities.

of the cells in *Sass6^{em5/em5}* Cetn2-eGFP blastocysts (33%) (*Figure 5C and D*). The data suggested that the mESC culture conditions are conducive to *de novo* centriole formation in the absence of SAS-6.

## The differentiation of *Sass6* mutant mESCs leads to centriole loss

To test whether the ability to form centrioles, albeit mostly abnormal, via a SAS-6-independent pathway is a characteristic of the pluripotent mESCs, we analyzed centriole formation in mESCs differentiated and enriched for neural progenitor cells (NPCs). As expected, centrosomes immunostained for TUBG and FOP were detected in almost all WT NPCs characterized by the expression of the intermediate filament NESTIN (97%, *Figure 5E and F*; *Figure 5—figure supplement 1A*). Notably, in *Sass6⁻/⁻* NPCs, the number of cells with centrosomes sharply decreased upon differentiation (from 56%, *Figure 3D and E*, down to 6%, *Figure 5E and F*). The data suggested that the SAS-6-independent centriole formation pathway is a property of pluripotent mESCs that is largely lost upon differentiation.

## Centriole formation in *Sass6⁻/⁻* mESCs relies on PLK4 activity

To understand the mechanism of how *Sass6⁻/⁻* mESCs are able to form centrioles in the absence of SAS-6, we tested the hypothesis that a higher concentration of centriolar components and a robust activity of PLK4 allow for centriole formation in *Sass6⁻/⁻* mESCs. We first examined whether the loss of centrioles upon mESCs differentiation correlated with a decrease in the recruitment of centriolar and centrosomal proteins that are important for the initial events of centriole assembly. Thus, we used immunostaining and quantified the centrosomal signal intensity of TUBG, CEP152, STIL, and SAS-4 in WT mESCs and NPCs (*Figure 6A–D*). Compared to mESCs, and in support of our hypothesis, NPCs showed highly reduced levels of all four investigated proteins (~ fourfold) (*Figure 6A–D*).

To functionally test whether SAS-6-independent centriole formation requires a potent activity of PLK4 in mESCs, we treated WT and *Sass6⁻/⁻* mESCs with different doses of the specific PLK4 inhibitor, centrinone B (*Wong et al., 2015*), and performed immunostaining for TUBG and FOP (*Figure 7A*). In WT mESCs treated with 100 nM centrinone B, the fraction of cells with centrosomes, as defined by the co-localization of TUBG and FOP, was not significantly different than in DMSO-treated control cells (82% and 90%, respectively) (*Figure 7A and B*). In contrast, the number of cells with centrosomes in *Sass6⁻/⁻* mESCs treated with 100 nM centrinone B was greatly reduced (6%) compared to DMSO-treated control cells (46%) (*Figure 7A and B*). In both WT and *Sass6⁻/⁻* mESCs treated with 500 nM centrinone B, centrosomes were identified only in a minor fraction of cells (10% and 4%,

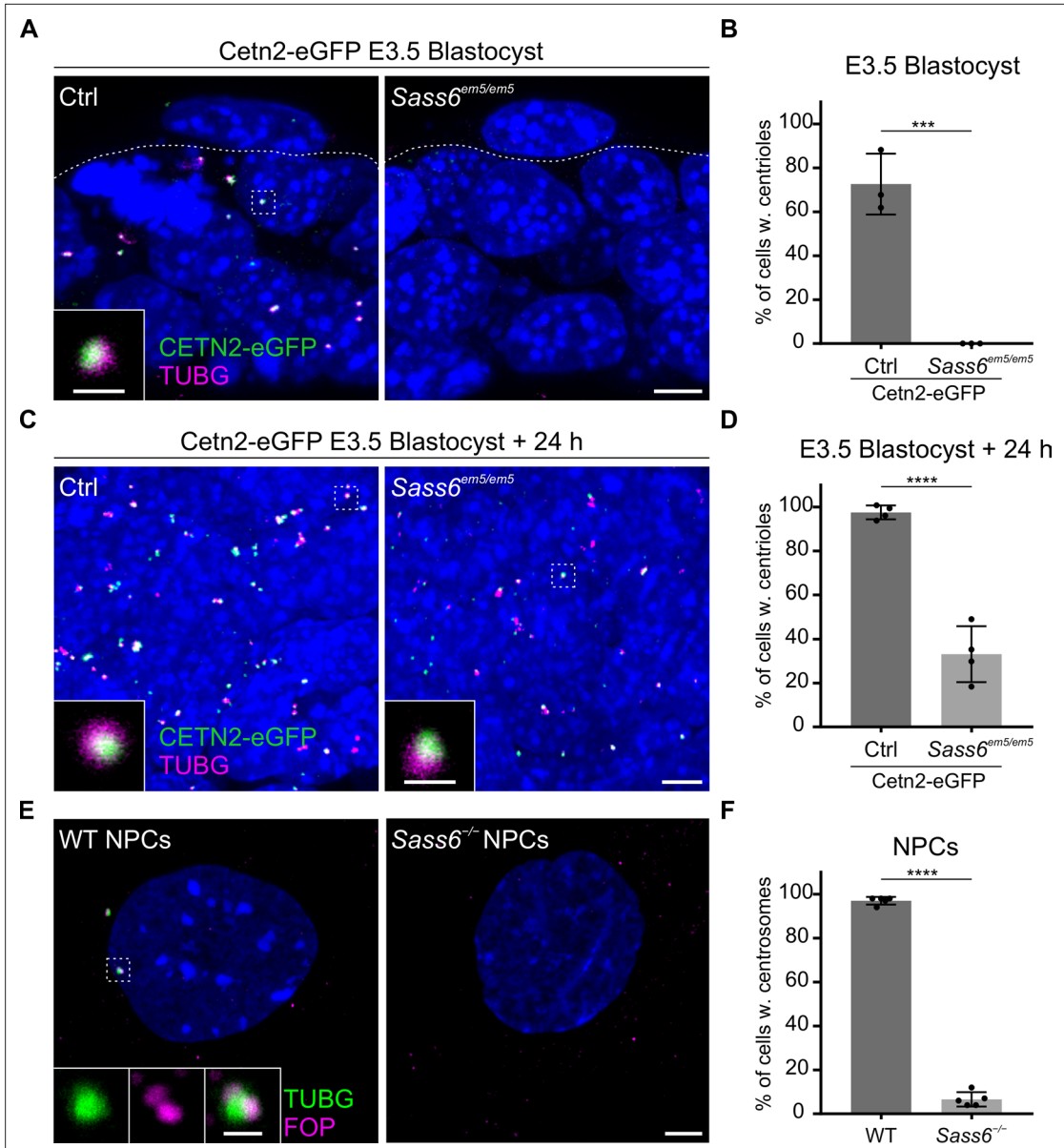

**Figure 5.** Centrioles in *Sass6* em5/em5 mouse embryonic stem cells (mESCs) are formed *de novo* during derivation from blastocysts and are lost upon differentiation. (**A**) Whole-mount immunostaining for TUBG on Cetn2-eGFP and *Sass6em5/em5* Cetn2-eGFP blastocysts at E3.5. Trophoblasts (top) and inner cell mass cells (bottom) are demarcated by the dashed line. The Inset is a magnification of the dashed square. Scale bars = 5 µm and 1 µm (inset). (**B**) Quantification of the percentage of cells with centrioles (TUBG and Centrin-eGFP) from E3.5 blastocysts in (**A**). Three blastocysts per genotype were used for the quantifications. Error bars represent mean ± SD WT: 73 ± 11% (n=200 cells); *Sass6em5/em5*: 0 ± 0% (n=175). ***p<0.001, (two-tailed Student's t-test). (**C**) Whole-mount immunostaining as mentioned in (**A**) on blastocysts after 24 hr in culture. (**D**) Quantification from (**C**) as mentioned in (**B**). Four blastocysts per genotype were used for the quantifications. WT: 98 ± 30% (n=630 cells); *Sass6em5/em5*: 33 ± 11% (n=690). ****p<0.0001. (**E**) Immunostaining for TUBG and FOP in WT and *Sass6−/−* cells after *in vitro* neural differentiation (NPCs). Insets are magnifications of the center of the dashed squares. Scale bars = 5 µm and 1 µm (insets). (**F**) Quantification of the percentage of cells with centrosomes (TUBG and FOP) in (**E**) from five independent experiments. Error bars represent mean ± SD WT: 97 ± 0% (n=1388 cells); *Sass6−/−*: 6 ± 0% (n=1068). ****p<0.0001, (two-tailed Student's t-test).

The online version of this article includes the following figure supplement(s) for figure 5:

**Figure supplement 1.** Wild-type (WT) and *Sass6−/−* mouse embryonic stem cells (mESCs) differentiated into neural progenitor cells (NPCs).

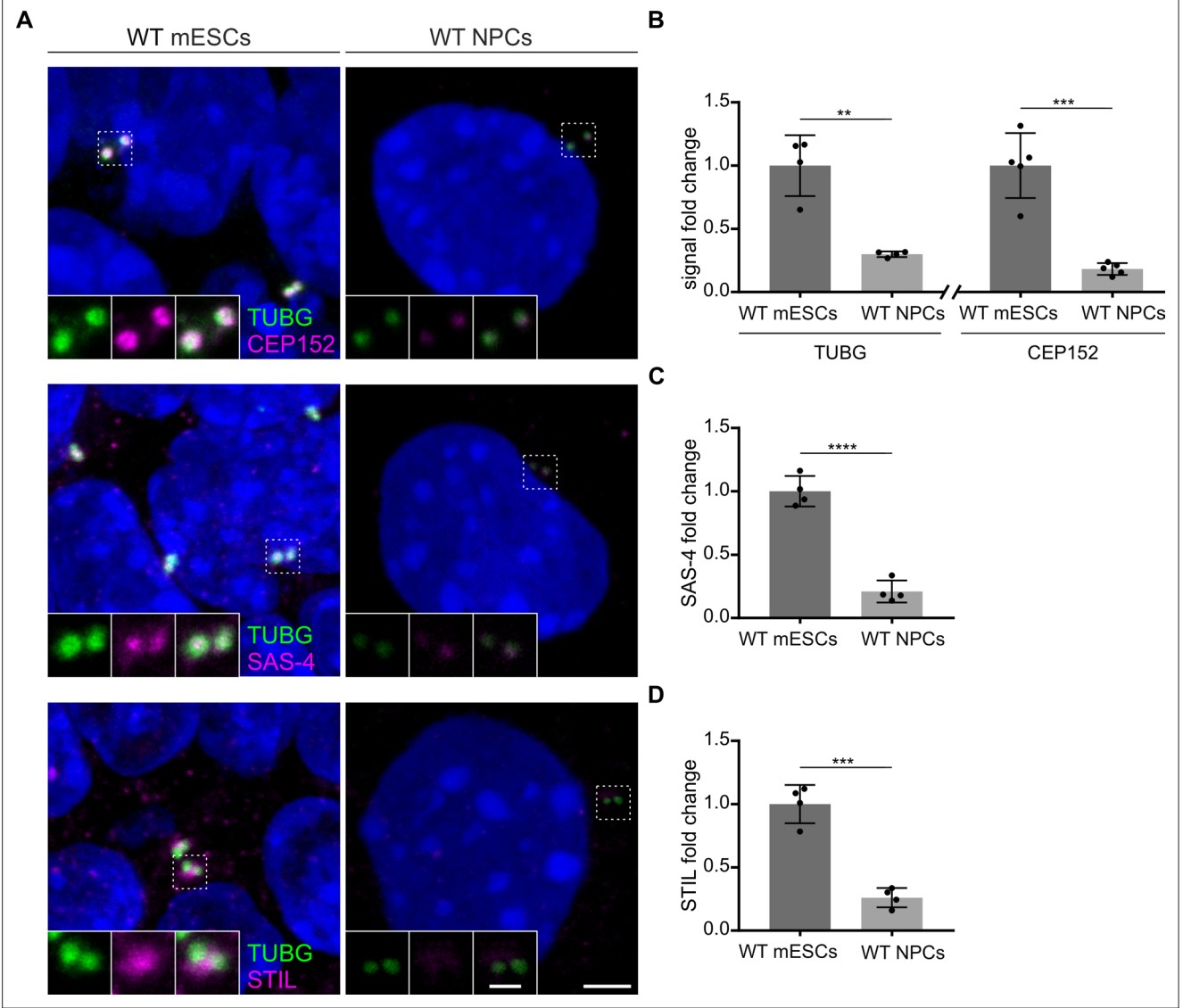

**Figure 6.** Levels of centrosomal components are reduced upon neural differentiation. (**A**) Immunostaining for TUBG and CEP152, TUBG and SAS-4, or TUBG and STIL in wild-type (WT) mouse embryonic stem cells (mESCs) and *in vitro* differentiated (neural progenitor cells, NPCs). Insets are magnifications of the center of the dashed squares. Scale bars = 3 μm and 1 μm (insets). (**B**) Quantification of the centrosomal TUBG and CEP152 signal from (**A**). Values were normalized to mESCs. Error bars represent mean ± s.d. Quantification of TUBG, mESCs: 1.00 ± 0.2 (n=1325 centrosomes from four independent experiments); NPCs: 0.03 ± 0.02% (n=789 from 4fourindependent experiments). Quantification of CEP152, mESCs: 1.00 ± 0.2 (n=1006 cells from five independent experiments); NPCs: 0.2 ± 0.04% (n=973 from five independent experiments). **p<0.01, ***p<0.001 (two-tailed Student's t-test). (**C**) Quantification of the centrosomal SAS-4 signal from (**A**) from four independent experiments. Values were normalized to mESCs. Error bars represent mean ± SD mESCs: 1.00 ± 0.1 (n=1297 centrosomes); NPCs: 0.2 ± 0.08% (n=790). ****p<0.0001, (two-tailed Student's t-test). (**D**) Quantification of the centrosomal STIL signal from (**A**) from four independent experiments. Values were normalized to mESCs. Error bars represent mean ± SD mESCs: 1.00 ± 0.13 (n=1132 centrosomes from four independent experiments); NPCs: 0.3 ± 0.07% (n=798). ***p<0.001, (two-tailed Student's t-test).

respectively) (*Figure 7A and B*). The data suggested that SAS-6-independent centriole formation in mESCs depends on high PLK4 activity.

## Discussion

In this study, we report that mutations in mouse *Sass6* cause embryonic arrest at mid-gestation with elevated levels of p53 and cell death, as well as the activation of the p53-, 53BP1-, and USP28-dependent mitotic surveillance pathway (*Figure 1*). We have previously reported similar phenotypes of elevated p53 and cell death for mutations in other genes essential for centriole duplication, such

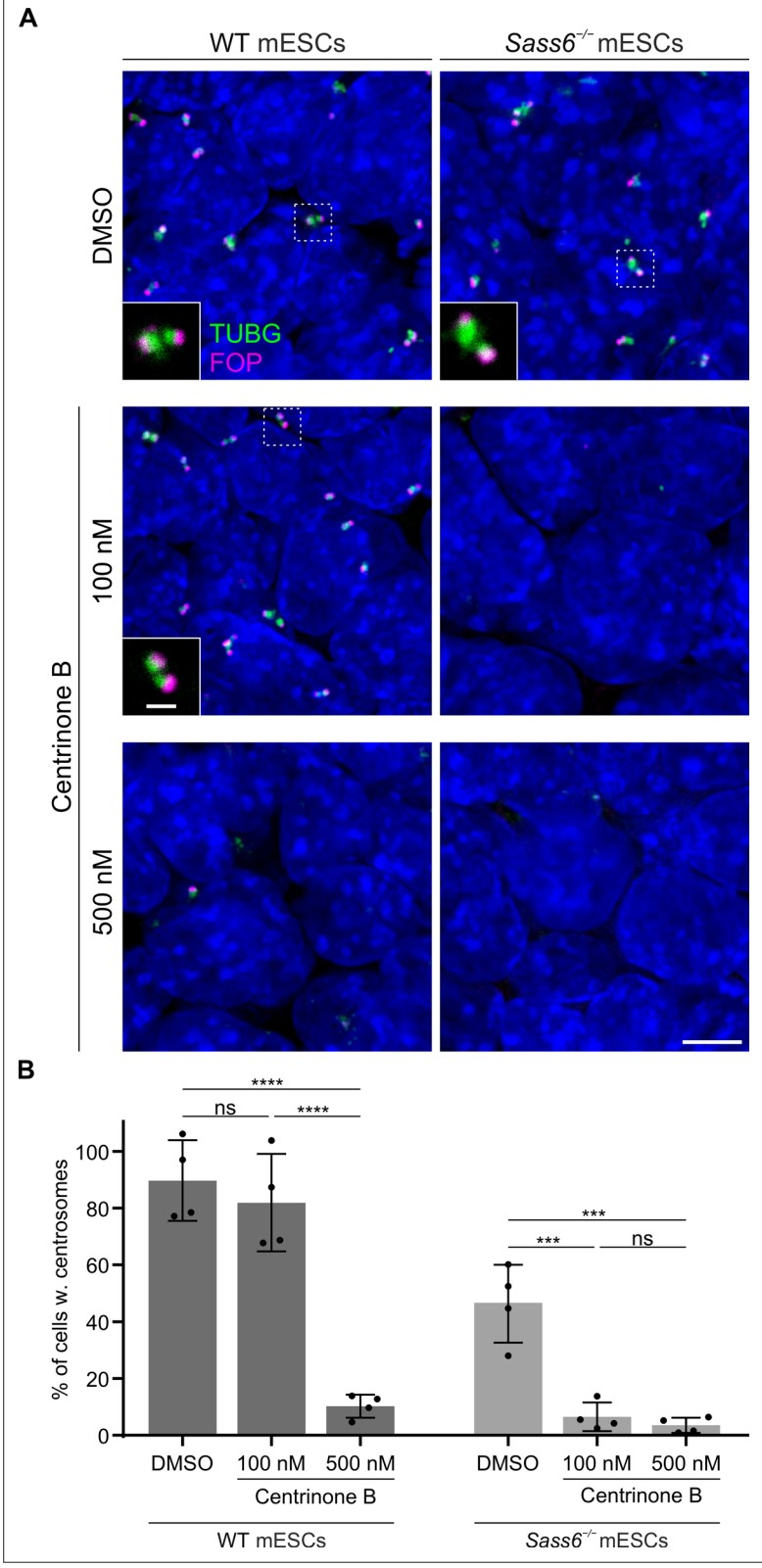

**Figure 7.** SAS-6-independent centriole formation in mouse embryonic stem cells (mESCs) depends on a threshold Polo-Like Kinase 4 (PLK4) activity. (**A**) Immunostaining for TUBG and FOP in wild-type (WT) and *Sass6⁻/⁻* mESCs treated for 4 days with DMSO, 100 nM or 500 nM centrinone B. Insets are magnifications of the center of the dashed squares and show the representative image of the majority of population. Scale bars = 5 μm and 1 μm

*Figure 7 continued on next page*

*Figure 7 continued*

(insets). (**B**) Quantification of the percentage of cells with centrosomes (TUBG and FOP) from (**A**) from four independent experiments. Error bars represent mean ± SD WT, DMSO: 90 ± 12% (n=5280 cells), 100 nM: 82 ± 15% (n=6083 cells), 500 nM: 10 ± 4% (n=4809 cells); *Sass6*[−/−], DMSO: 46 ± 12% (n=5786 cells), 100 nM: 6 ± 4% (n=7502 cells), 500 nM: 4 ± 2% (n=6220 cells). ***p<0.001, ****p<0.0001, ns = not significant with p>0.05 (one-way ANOVA with Tukey's multiple comparisons).

as *Cenpj* and *Cep152* (*Bazzi and Anderson, 2014a*). The current data demonstrated that mouse SAS-6 is required for centriole formation in developing mouse embryos (*Figure 2*), as expected from the established role of its orthologs in *C. elegans* and human cells (*Gupta et al., 2020*; *Leidel et al., 2005*; *Wang et al., 2015*). Together, the data provide further support that the activation of the mitotic surveillance pathway is not specific to the loss of specific centriolar proteins but rather the loss of the centriole/centrosome structure and function *per se* (*Bazzi and Anderson, 2014b*).

To our surprise, and in contrast to the mouse embryo, removing SAS-6 from mESCs is still compatible with the formation of centrioles, but these centrioles are mostly abnormal and defective (*Figure 3*). Our data support the published literature on SAS-6-independent centriole duplication that also leads to the formation of abnormal centrioles in evolutionarily more distant organisms, namely

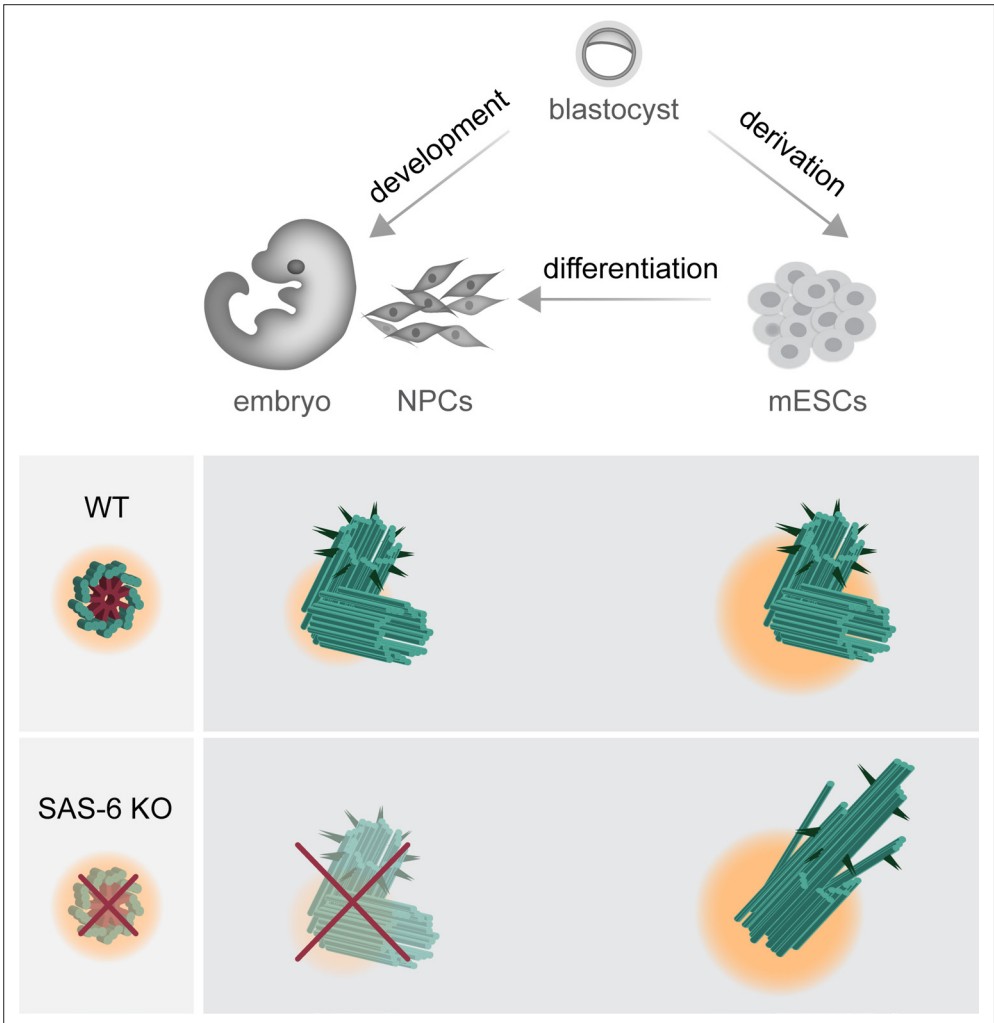

**Figure 8.** Graphical model depicting the consequences of SAS-6 loss in mouse embryos, mouse embryonic stem cells (mESCs), and neural progenitor cells (NPCs). Compared to mouse embryos and *in vitro* differentiated NPCs, mESCs exhibit a higher concentration of centrosomal components and a robust Polo-Like Kinase 4 (PLK4) activity, as indicated by changes in pericentriolar material color and size. This difference permits the formation of abnormal centrioles in *Sass6*[−/−] mESCs, while it results in the loss of centrioles in developing mouse embryos and NPCs.

*C. Reinhardtii* and *D. melanogaster* (*Nakazawa et al., 2007*; *Rodrigues-Martins et al., 2007*). Similar phenotypes of abnormal centrioles were also described in human RPE-1 cells lacking the SAS-6 oligo-merization domain; however, the loss of the entire SAS-6 protein in these cells leads to the loss of centrioles (*Wang et al., 2015*). Interestingly, we demonstrate that centrioles appear during the deriva-tion process of SAS-6-deficient mESCs but are lost again upon differentiation (*Figure 5*). The findings extend the observations on abnormal centrioles or centriole loss upon SAS-6 depletion and reveal the differential requirements for SAS-6 even within cells of the same species (*Figure 8*).

In *D. melanogaster*, *C. elegans,* and human cells, SAS-6 has been shown to directly interact with a downstream protein STIL, and provides a structural basis for the recruitment of other centriole dupli-cation proteins, such as CEP135 and SAS-4 (*Lin et al., 2013*; *Qiao et al., 2012*; *Tang et al., 2011*). The loss of STIL or SAS-4 leads to centriole duplication failure in all organisms and cell types studied to date (*Bazzi and Anderson, 2014a*; *Vulprecht et al., 2012*). Notably, unlike SAS-6, SAS-4 is essential for centriole formation in mESCs (*Xiao et al., 2021*). Our data show that STIL localized to half of the centrosomes in *Sass6*-null mESCs, which might account for centriole duplication failure in almost half of the centrosomes that contain only single centrioles. Our data also suggest that pluripotent mESCs without SAS-6 have a bypass pathway of SAS-4 recruitment. Given that centrioles in mESCs show higher levels of centrosomal components compared to NPCs (*Figure 6*), and that SAS-6-independent centriole formation in mESCs depends on high PLK4 activity (*Figure 7*), we propose that these factors drive SAS-6-independent centriole formation by supporting the stability of centriole intermediate structures in mESCs, while embryos lacking such a mechanism experience rapid disassembly of the intermediate assemblies.

Although centrioles are still present in *Sass6*$^{-/-}$ mESCs, they exhibit profound proximal and distal defects. Whether this phenotype arises as a consequence of an improper initiation of assembly or a later destabilization of the microtubule triplets, are still open questions. The cartwheel protein CEP135 has been shown to be important for centriole stability in *T. thermophila* and human cells (*Bayless et al., 2012*; *Lin et al., 2013*). Our data show that centrioles in *Sass6*$^{-/-}$ mESCs exhibit abnormal localization of CEP135 (*Figure 4—figure supplement 1B, C*). We speculate that the lack of an initial stable cartwheel scaffold that provides a ninefold symmetry template and the ensuing mis-localization of CEP135 may account for the abnormal centriole architecture and its instability.

In conclusion, our work provides new fundamental insights into alternative and SAS-6-independent pathways of centriole formation in mammalian cells. It also highlights that mESCs are a special *in vitro* model for centriole biology that can tolerate centriolar aberrations, such as in *Sass6* mutants, or even the loss of centrioles, as in *Cenpj* mutants, without undergoing apoptosis or cell cycle arrest (*Lambrus and Holland, 2017*; *Xiao et al., 2021*). The difference in centrosome biology between mouse embryos and mESCs adds to a growing body of evidence of variable centrosome composition and function among different cell types (*O'Neill et al., 2022*; *Xie et al., 2021*).

## Materials and methods
### Mice and genotyping

The following mouse alleles for *Usp28*$^{+/-}$ (*Usp28*$^{em1/Baz}$) and *Trp53bp1*$^{+/-}$ (*Trp53bp1*$^{em1/Baz}$) (*Xiao et al., 2021*), as well as Cetn2-eGFP (Tg$^{(CAG-EGFP/CETN2)3-4Jgg}$, as hemizygous) (*Bangs et al., 2015*; *Higginbotham et al., 2004*) were used in this study. The *Trp53*$^{+/-}$ null allele was generated by crossing *Trp53*$^{+/tm1.1Brn}$ (Jax stock no. 008462), +/- mice with K14-Cre (*Hafner et al., 2004*) females, which express the Cre recombinase in the zygote. *Sass6*$^{+/em4}$ (*Sass6*$^{em4/Baz}$, in exon 4) and *Sass6*$^{+/em5}$ (*Sass6*$^{em5/Baz}$, in exon 5) mice were generated using CRISPR/Cas9 genome-editing by the CECAD in vivo Research Facility (ivRF, Branko Zevnik) (*Table 1*), where gRNA, Cas9 protein, and mRNA were delivered to fertilized zygotes by pronuclear injection or electroporation (*Chu et al., 2016*; *Tröder et al., 2018*). The animals were housed and bred under SOPF conditions in the CECAD animal facility. The animal generation application (84–02.04.2014 .A372), notifications and breeding applications (84–02.05.50.15.039, 84–02.04.2015 .A405, UniKöln_Anzeige§4.20.026, 84–02.04.2018 .A401, 81–02.04.2021 .A130) were approved by the Landesamt für Natur, Umwelt, und Verbraucherschutz Nordrhein-Westfalen (LANUV-NRW) in Germany. The phenotypes were analyzed in the FVB/NRj background. Genotyping was carried out using standard and published PCR protocols as cited or described in this work. For the new *Sass6*$^{+/em4}$ and *Sass6*$^{+/em5}$ mouse alleles, the PCR products (primers shown in ***Supplementary***

*file 1*) were digested with *AvaII* and *HpyCH4IV* restriction enzymes (New England BioLabs; Ipswich, MA, USA), respectively, to distinguish between the WT and mutant alleles. *AvaII* cut the product in the *Sass6^{+/em4}* mutant allele, whereas *HpyCH4IV* did not cut the product in the *Sass6^{+/em5}* mutant allele.

## Mouse embryonic stem cell culture and centrinone B treatment

mESCs were derived from male WT, *Sass6^{+/em5}* Cetn2-eGFP *or Sass6^{em5/em5}* Cetn2-eGFP blastocysts, mostly FVB/NRj strain, as previously described and maintained at 37 °C with 6% $CO_2$ (*Bryja et al., 2006*). The established mESC lines were adapted to feeder-free conditions and cultured on 0.1% gelatin (PAN Biotech, Aidenbach, Germany) coated plates in Knock-Out DMEM (Thermo Fisher Scientific; Waltham, MA, USA) supplemented with 15% HyClone fetal bovine serum (FBS; VWR; Radnor, PA, USA), 2 mM L-glutamine (Biochrom; Berlin, Germany), 1% penicillin/streptomycin (Biochrom), 0.1 mM MEM non-essential amino acids (Thermo Fisher Scientific), 1 mM sodium pyruvate (Thermo Fisher Scientific), 0.1 mM β-mercaptoethanol (Thermo Fisher Scientific), 1000 U/ml leukemia inhibitory factor (LIF; Merck; Darmstadt, Germany), and with 1 µM PD0325901 (Miltenyi Biotec; Bergisch Gladbach, Germany) and 3 µM CHIR99021 (Miltenyi Biotec). For the centrinone B experiment, mESCs were cultured for four days in 100 nM or 500 nM centrinone B (MedChemExpress, Monmouth Junction, NJ, USA) in mESCs culture media.

## Generation of *Sass6* mutant mESCs using CRISPR/Cas9

For the generation of the CRISPR/Cas9-mediated *Sass6* knockout mESCs line, a pair of gRNAs targeting the 5′ and 3′ ends of the entire *Sass6* ORF (*Figure 2—figure supplement 1* and *Table 2*) were cloned as double-stranded oligo DNA into BbsI and SapI sites in pX330-U6-Chimeric_BB-CBh-hSpCas9 vector (Addgene; Watertown, MA, USA) modified with a Puro-T2K-GFP cassette containing puromycin-resistance by Dr. Leo Kurian's research group (Center for Molecular Medicine Cologne). mESCs in suspension were transfected with the modified pX330 vector containing the pair of gRNAs using Lipofectamine 3000 (Thermo Fisher Scientific). The cells were then subjected to selection using puromycin (2 µg/ml, Sigma-Aldrich; St. Louis, MO, USA) 24 hr post-transfection for 2 days. After recovery for 5 days, individual colonies were picked and screened for the *Sass6* locus deletion by PCR (*Figure 3A* and *Table 2*; *Supplementary file 1*). The cells were used for experiments after four passages.

## RT-PCR

RNA was extracted from mESCs using the RNeasy Plus Mini Kit (Qiagen; Hilden, Germany). Reverse transcription was performed using SuperScript III reverse transcriptase (Invitrogen; Waltham, MA, USA) and oligo(dT) primer per the manufacturer's recommendation. The cDNA was used for PCR analysis using the primers shown in *Supplementary file 1*.

## Embryo dissection, immunofluorescence staining, and image acquisition

Timed pregnant female mice were sacrificed by cervical dislocation. Post-implantation embryos (E7.5-E9.5) were dissected in ice-cold PBS with 0.1% Tween20 (AppliChem; Darmstadt, Germany), and fixed overnight in 2% paraformaldehyde (PFA; Carl Roth; Karlsruhe, Germany) at 4 °C, or in methanol for 30 min at –20 °C for Ultrastructure-Expansion Microscopy (U-ExM) and SAS-6 immunostaining. Embryos were dehydrated overnight at 4 °C in 30% sucrose and embedded in O.C.T. (Sakura Finetek; Alphen an den Rijn, Netherlands) for cryo-sectioning at 8 µm sections using a CM1850 Cryostat (Leica Biosystems; Wetzlar, Germany).

Pre-implantation E3.5 blastocysts were recovered by flushing the uterine horns with EmbryoMax M2 Medium (Sigma-Aldrich), and fixed in 4% PFA for 20 min at room temperature (RT) and 20 min at 4 °C. Blastocysts were then permeabilized for 3 min with 0.5% Triton X-100 (Sigma-Aldrich) in PBS and three times for 10 min with immunofluorescence (IF) buffer containing 0.2% Triton X-100 in PBS. After blocking with 10% heat-inactivated goat serum in IF buffer, the blastocysts were incubated overnight with the primary antibodies at 4 °C, followed by a 2 hr incubation with the secondary antibodies and DAPI at RT (1:1000, AppliChem). Blastocysts were imaged in single drops of PBS covered with mineral oil, followed by genotyping.

For IF staining of embryo sections from the brachial region (forelimb and heart level), the slides were post-fixed in methanol for 10 min at –20 °C, washed with IF buffer, and blocked with 5%

**Table 3.** List of primary antibodies used in this study.

| Antigen | Company | Catalog number | Dilution |
|---|---|---|---|
| Ac-TUB | Sigma-Aldrich | T6793 | IF (1:1000)<br>U-ExM (1:500) |
| α-TUB | Sigma-Aldrich | T6074 | U-ExM (1:500) |
| β-TUB | Sigma-Aldrich | T8328 | U-ExM (1:200) |
| ARL13B | Proteintech | 17711–1–AP | IF (1:1000) |
| CEP97 | Proteintech | 22050–1–AP | U-ExM (1:50) |
| CEP135 | Proteintech | 24428–1–AP | U-ExM (1:200) |
| CEP152 | Sigma-Aldrich | HPA039408 | IF (1:100) |
| CEP164 | Proteintech | 22227–1–AP | IF (1:1000)<br>U-ExM (1:500) |
| Cl. CASP3 | Cell Signaling | 9661 | IF (1:400) |
| CP110 | Proteintech | 12780–1–AP | U-ExM (1:250) |
| FOP | Sigma-Aldrich | WH0011116M1 | IF (1:500) |
| NESTIN | BioLegend<br>R&D Systems | PRB-315C<br>MAB2736 | IF (1:500)<br>IF (1:300) |
| GAPDH | Merck | CB1001 | WB (1:10,000) |
| p53 | Cell Signaling | 2524 | IF (1:2000) |
| POC5 | Bethyl Laboratories | A303-341A | U-ExM (1:250) |
| SAS-4 | A kind gift from Pierre Gönczy, Ecole Polytechnique Fédérale de Lausanne (EPFL), Lausanne, Switzerland | | IF (1:500) |
| SAS-6 | A kind gift from Renata Basto, Institut Curie, Paris, France | | IF (1:300)<br>WB (1:1000) |
| STIL | Bethyl Laboratories | A302-441A | IF (1: 400)<br>U-ExM (1:200) |
| TUBG | Sigma-Aldrich | T6557 | IF (1:1000) |

heat-inactivated goat serum in IF buffer. The slides were incubated overnight with primary antibodies at 4 °C followed by 1 hr incubation with secondary antibodies and DAPI at RT (1:1000, AppliChem), then mounted with ProLong Gold Antifade reagent (Cell Signaling Technology; Danvers, MA, USA).

For IF staining of mESCs, the cells were cultured in Lab-Tek II chamber slides or ibiTreat μ slides (*Figure 6*), coated with 0.1% gelatin, fixed with 4% PFA for 10 min at RT, and post-fixed with methanol for 10 min at –20 °C. The cells were then permeabilized for 5 min using 0.5% Triton X-100 in PBS. After blocking with IF buffer with 5% heat-inactivated goat serum, the cells were incubated overnight with the primary antibodies at 4 °C, followed by 1 hr incubation with secondary antibodies and DAPI (1:1000, AppliChem). The slides were mounted with ProLong Gold (Cell Signaling Technology). Images were obtained using TCS SP8 (Leica Microsystems) confocal microscope with PL Apo 63 x/1.40 Oil CS2 objective and Stellaris 5 (Leica Microsystems) confocal microscope with HC PL APO 63 x/1.30 GLYC CORR CS2 objective.

## Antibodies

All primary antibodies are listed in *Table 3*. The following secondary antibodies were used: Alexa Fluor 488, 568, or 647 conjugates (Life Technologies) (IF 1:1000, U-ExM 1:400).

## Ultrastructure-expansion microscopy (U-ExM)

For U-ExM of mESCs, the cells were cultured on 12 mm glass cover slips (VWR) coated with 0.1% gelatin and fixed with methanol for 10 min at –20 °C. For U-ExM of embryos, cryo-sections were collected on 12 mm glass coverslips, O.C.T. was washed away in PBS. Sample expansion was performed as

described in *Gambarotto et al., 2019*. Briefly, the cover slips were incubated in 1.4% formaldehyde (Sigma-Aldrich), 2% acrylamide (Sigma-Aldrich) in PBS for 5 hr at 37 °C. Gelation was carried out in 35 μl of monomer solution 23% (w/v) sodium acrylate (Sigma-Aldrich), 10% (w/v) acrylamide, 0.1% (w/v) N,N'-methylenebisacrylamid (Sigma-Aldrich) in 1x PBS supplemented with 0.5% APS (Bio-Rad; Feld-kirchen, Germany) and 0.5% TEMED (Bio-Rad) for 5 min on ice and 1 h at 37 °C, followed by gel incubation in denaturation buffer 200 mM sodium dodecyl sulfate (SDS; AppliChem), 200 mM NaCl and 50 mM Tris in $H_2O$ (pH 9) for 1.5 hr at 95 °C, and initial overnight expansion in dd$H_2O$ at RT. The gels were incubated with primary antibodies for 3 hr at 37 °C on an orbital shaker, then with secondary antibodies for 2.5 hr at 37 °C. After the final overnight expansion in dd$H_2O$ at RT, the expanded gel size was accurately measured using a caliper, and then mounted on 12 mm glass cover slip coated with Poly-D-Lysine (Thermo Fisher Scientific) and imaged using a TCS SP8 confocal microscope with a Lightning suite (Leica Microsystems) to generate deconvolved images.

## Blastocyst *in vitro* culture

Blastocysts at E3.5 were cultured for 24 hr at 37 °C and 6% $CO_2$, in ibiTreat μ-Slides (Ibidi GmbH, Munich, Germany) on feeder cells that were previously growth-arrested with a 2 hr mitomycin C treatment (10 μg/ml, Sigma-Aldrich), in mESCs derivation media mESCs media except with the FBS replaced with Knockout Serum Replacement (15%, Thermo Fisher Scientific). The cultures were fixed and processed for IF staining as described above for the E3.5 blastocysts. Slides were mounted with VectaShield mounting medium (Linaris; Dossenheim, Germany).

## Neural differentiation of mESCs

For neural differentiation, embryoid bodies were generated using the hanging drop method (*Wang and Yang, 2008*). Around 1000 mESCs were suspended in 20 μl of differentiation medium mESCs media without LIF, PD0325901, and CHIR99021, supplemented with 1 μM retinoic acid (Sigma-Aldrich). After 3 days, the resulting embryoid bodies were plated on ibiTreat μ-slides coated with fibronectin (30 μg/ml, Sigma-Aldrich) and cultured for 4 days. The cells were fixed in methanol for 10 min at –20 °C for centrosomal staining or 4% PFA for 10 min at RT for other stainings, and processed for IF similar to mESCs. Slides were mounted with VectaShield mounting medium (Linaris).

## Western blotting

Western blots were performed according to standard procedures (*Mahmood and Yang, 2012*). Briefly, dissected embryos at E9.5 were lysed in Laemmli buffer, and the cells were lysed in RIPA buffer 150 mM NaCl, 50 mM Tris pH 7.6, 1% Triton X-100, 0.25% sodium deoxycholate, and 0.1% SDS (AppliChem) with an ethylenediaminetetraacetic acid (EDTA)-free protease inhibitor cocktail (Merck), phosphatase inhibitor cocktail sets II and IV (Merck), and phenylmethylsulfonyl fluoride (PMSF; Sigma-Aldrich). 80 μg of protein per sample was used for SDS-PAGE. Samples were then blotted onto poly-vinylidene difluoride membrane (Merck). After blocking in 5% non-fat milk (Carl Roth), the membrane was incubated overnight at 4 °C with a SAS-6 or GAPDH antibody. Secondary antibodies coupled to horseradish peroxidase were used for enhanced chemiluminescence signal detection with ECL Prime Western Blotting System (GE Healthcare).

## Image analysis

For signal quantification using ImageJ (NIH), the signal intensity from the nuclear area, as determined by DAPI staining, was normalized to DAPI intensity. The signal intensity from the centrosomal area was determined by TUBG staining. The fold change of p53 or Cl. CASP3 was defined as a ratio between normalized signal intensity to the mean signal intensity of the WT from all replicates, and the fold change of centrosomal proteins was defined as a ratio between mean signal intensity to the mean signal intensity of the mESCs from all replicates. The percentage of cells with centrosomes in embryos and mESCs was defined as a ratio between centrosome number manually quantified using ImageJ and the number of nuclei quantified using the IMARIS software (Bitplane; Belfast, United Kingdom). Centriole length was measured using ImageJ from nearly parallel-oriented centriole walls stained for TUB. The length was corrected for the expansion factor obtained from dividing the gel size after expansion by the size of a cover slip used for gelation (12 mm).

## Statistical analysis

To identify statistical differences between two or more groups, two-tailed Student's t-test or one-way ANOVA with Tukey's multiple comparisons was performed. $p < 0.05$ was used as the cutoff for significance. The statistical analyses were performed using Microsoft Excel (Microsoft Corporation, Redmond, WA, USA) or Prism (GraphPad; San Diego, CA, USA) and the graphs were generated using Prism.

## Acknowledgements

We thank the CECAD in vivo research facility (Branko Zevnik) for generating and maintaining our mouse lines and the CECAD imaging facility for microscopy support. We thank Charlotte Meyer-Gerards for assistance with generating the schematics. We are grateful to members of the lab and colleagues for the critical reading of the manuscript. The work was funded by the Deutsche Forschungsgemeinschaft (DFG, German Research Foundation) to H.B - Project-ID 73111208 - SFB829 'Molecular Mechanisms regulating Skin Homeostasis,' subproject A12; and - Project-ID 331249414 - BA 5810/1-1. The funders had no role in study design, data collection and analysis, decision to publish, or preparation of the manuscript.

## Additional information

### Funding

| Funder | Grant reference number | Author |
| --- | --- | --- |
| Deutsche Forschungsgemeinschaft | 73111208 - SFB829 | Hisham Bazzi |
| Deutsche Forschungsgemeinschaft | 331249414 - BA 5810/1-1 | Hisham Bazzi |

The funders had no role in study design, data collection and interpretation, or the decision to submit the work for publication.

### Author contributions

Marta Grzonka, Conceptualization, Software, Formal analysis, Investigation, Visualization, Methodology, Writing – original draft, Writing – review and editing; Hisham Bazzi, Conceptualization, Supervision, Funding acquisition, Investigation, Writing – original draft, Project administration, Writing – review and editing

### Author ORCIDs

Marta Grzonka (iD) http://orcid.org/0000-0003-3389-4816
Hisham Bazzi (iD) https://orcid.org/0000-0001-8388-4005

### Ethics

The animal generation application (84-02.04.2014.A372), notifications and breeding applications (84-02.05.50.15.039, 84-02.04.2015.A405, UniKöln_Anzeige§4.20.026, 84-02.04.2018.A401, 81-02.04.2021.A130) were approved by the Landesamt für Natur, Umwelt, und Verbraucherschutz Nordrhein-Wesdalen (LANUV-NRW) in Germany.

### Decision letter and Author response

Decision letter https://doi.org/10.7554/eLife.94694.sa1
Author response https://doi.org/10.7554/eLife.94694.sa2

## Additional files

### Supplementary files

- Supplementary file 1. List of primers.
- MDAR checklist

## Data availability

All data generated or analysed during this study are included in the manuscript and supporting files; source data files have been provided for *Figures 2 and 3*.

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
