## [Editor Report]

This is an important study on the formation of mouse centrioles in the embryo and stem cells derived from them. The authors provide convincing evidence on the unique role of the SAS-6 protein and the entire cartwheel complex highlighting a mechanism that suggests a specific cell cycle related function of centriole formation and its maintenance.

---

## [Decision Letter]

[Editors' note: this paper was reviewed by Review Commons.]

---

## [Author Response]

General Statements

We would like to thank the editor for handling our manuscript entitled, “Mouse SAS‑6 is required for centriole formation in embryos and integrity in embryonic stem cells”, and the reviewers for the insightful comments and suggestions to improve our work.

We have fully revised our manuscript per the reviewers’ suggestions and added significantly to:

1) the analyses of the *Sass6* mutant embryos *in vivo*

2) the molecular mechanism of how *Sass6* mutant mESCs in vitro can form centrioles

In our opinion, the novelty of our work is clearly evident in the discovery that mESCs lacking SAS-6 (gene named *Sass6*) are still able to form centrioles, albeit mostly abnormal, which is also shared by the reviewers. This is in contrast to *Cenpj* (*Sas-4*) mutant mESCs for example, which lack centrioles (Xiao*, Grzonka* et al., EMBO Reports 2021), and cultured human cell lines without SAS-6, which have been shown to lose centrosomes. Below, we provide a point-by-point response to the reviewers.

Description of the revisions that have already been incorporated in the transferred manuscript:Reviewer #1 (Evidence, reproducibility and clarity (Required)):The article by M. Grzonka and H. Bazzi entitled: Mouse Sas-6 is required for centriole formation in embryos and integrity in embryonic stem cells, describes new findings in novel mouse models of Sas-6 knockouts (KO). This is an interesting study that reports two different mouse Sas6 KO models and the depletion of Sas6 from mouse embryonic stem cells (mESCs). This type of analysis has never been done before and so it reveals and describes a role for Sas-6 in centriole biogenesis in mouse.

We thank the reviewer for highlighting the novelty of our work on the roles of SAS-6 in mice.

The authors compare their analysis with Sas-4 KO and overall found similar results when compared to previous work from H. Bazzi, when Sas-4 was depleted in mouse embryos. Due to the mitotic stopwatch pathway, Sas6KO embryos die during development at extremely early stages and this can be rescued by depletion in p53 and other members of the pathway.Perhaps, not so surprisingly, these embryos do not contain centrioles, showing that in vivo, Sas6 is absolutely required for centriole duplication. More surprisingly, however, in cultures of mESCs, established and propagated in vitro, Sas-6 crispr induced KO, does not result in lack of centrioles. Instead, abnormal structures that show aberrant morphologies, length, and incapacity to assemble cilia were detected. In principle, this means that centrioles can be assembled independently of Sas-6, even if not in the correct manner.

We again thank the reviewer for astutely pointing out the most surprising finding in our data, which is that mESCs lacking SAS-6 can still form centrioles.

The authors interpret these differences as possible differences in the pathways involved in centriole assembly and propose different requirements in different cell types, within the same species.I have problems with this interpretation. To me is very difficult to understand, how the "protein" absolutely required for cartwheel assembly at the early stages of centriole biogenesis, can be essential and dispensable at the same time. Although, I may be wrong, I think the authors have not envisage other possibilities to interpret their data, which should be taken into consideration.

We agree with the reviewer that SAS-6 is currently considered in the centrosome field as one of the “core” centriole formation or duplication factors and that it is a major component of the cartwheel scaffold during the early phase of centriole biogenesis. Although, the absence of centrioles in the *Sass6* mutant mouse embryos *in vivo* supports the essential function of SAS-6, and perhaps the cartwheel, in centriole formation; the mere presence of centrioles in mESCs indicates that SAS-6, and again the cartwheel, is not essential for the existence of centrioles in these cells. Because this is a major finding that we would like to bring across from our study, we better highlighted and clarified it in the new version of the manuscript as described below. In fact, in points #4 and #5, we share the same possible explanation for the difference in the phenotypes between *Sass6* mutant mouse embryos and mESCs as the reviewer and we have included them in the revised discussion.

1) I do not know anything about ESC and ESC cultures. So maybe this is a stupid suggestion. But can't they be derived exactly from the same genetic background of SAS-6KO embryos? Because the way the two (or even 3 as there are 2 mouse KO lines) are generated is different. Why is that?

The reviewer is correctly suggesting that the mESCs can be derived from the *Sass6* mutant blastocysts. We would also like to direct the attention of the reviewer that we have cultured the blastocysts (E3.5) from the *Sass6^em5/em5^* mutants, which as we show lack centrioles at E3.5, and the cells indeed start to form centrioles just 24 h post-culture (Figure 5A-D). Given the literature on the essential role for SAS-6 in centriole formation in other cell types, we also generated a more convincing *Sass6^-/-^* null allele in mESCs by deleting the entire ORF of *Sass6* (more on this point below).

As suggested by the reviewer and to build on these findings, we generated and propagated a mESC line from the *Sass6^em5/em5^* mutants, together with a Cetn2-eGFP transgene to mark centrioles (Figure 3—figure supplement 1B-F). These *Sass6^em5/em5^* Cetn2-eGFP–derived mESCs show CETN2-eGFP-positive centrioles, which are largely similar in number and architecture to those in the CRISPR-generated *Sass6^-/-^* null mESCs (Figure 3—figure supplement 1B-F compared to Figure 3D-H).

2) Still on mESCs, are the authors sure that there are no WT Sas-6 mRNAs still present in their ESC cells? Because tiny amounts are maybe sufficient to allow the initial cartwheel structure. In FigS2B, I can see a really faint band, very faint but it is there.

Due to the nature of the surprising finding that *Sass6* mutant mESCs can still form centrioles, we understand the concerns and suggestions of this reviewer and the other reviewers in this regard. We have generated several *Sass6* mutant alleles in mESCs (in exons 2, 4 or 5), in which we used Western blots to check whether they were null alleles or not. We used different commercial (Proteintech cat# 21377-1-AP, Sigma-Aldrich cat# HPA028187 and Santa Cruz cat# SC-81431) and non-commercial (kind gift from Renata Basto, Institute Curie) antibodies. The SAS-6 antibody from the Basto lab gave the most reliable and reproducible results. Using this antibody, and in our own interpretation, we were not able to detect SAS-6 by Western blots in *Sass6* mutant mESCs. We concluded that SAS-6 in mESCs (and mouse embryos, see below) is expressed at low levels.

Surprisingly, we always detected centrioles in the different *Sass6* mutant mESCs, even those derived from the *Sass6^em5/em5^* Cetn2-eGFP mutant blastocysts (Figure 3—figure supplement 1B-F), which in the blastocysts had no detectable centrioles (Figure 5A).

For a more definitive knockout in mESCs, we decided to bi-allelically delete the entire *Sass6* ORF DNA from the ATG to the TAA (over 34 Kb of DNA, Figure 3A). According to the central dogma of molecular biology, when there is no DNA, then there should be no mRNA (Figure 3B) and no protein (Figure 3C, Figure 3—figure supplement 1A). In confirmation of this premise, RT-PCR data showed that *Sass6* mRNA is not detectable in these *Sass6^-/-^* null mESCs (Figure 3B). Also, Western Blot analyses (Figure 3C) and immunofluorescence analyses (Figure 3—figure supplement 1A) did not detect SAS-6 in these cells.

These *Sass6^-/-^* mESCs started from a single cell and have been passaged more than 30-40 times without losing centrioles. This is how knockouts have been and are produced. If this mutant is still not convincing, then we respectfully ask that the reviewers provide their own suggestion on what will be more convincing. In our humble opinion, this *Sass6^-/-^* mESCs line can be used to test the specificity of the antibodies in mouse cells and not the other way around.

3) This last point goes also with the western-blot of Figure S2C- there is still a band, very tiny between the two very tick bands (marked with *). Maybe separating proteins better will help visualizing the real Sas-6 band? Have they used the Sas6 ab in other WBs from the KO embryos, for example? Can they use the Sas6 ab in immunostaining to show if the assembled abnormal centrioles completely lack Sas6. This will allow to distinguish between the hypothesis of having some (even if not much) sas6 left?

The answer to these questions is above in point #2. In addition, we have used the Basto lab antibody for SAS-6 for Western blots on mouse embryos, which detect low levels of SAS-6 in controls and no signal in the mutants (Figure 2B). We also repeated the SAS-6 Western blots on mESCs and achieved better band resolution (Figure 3C). Using this antibody for immunofluorescence showed that the *Sass6^em4/em4^* mutant is hypomorphic (Figure 2C, D), whereas the *Sass6^em5/em5^* mutant showed very low, most likely background, staining (Figure 2C, D). For mESCs, we decided to delete the entire *Sass6* ORF DNA and generate homozygous *Sass6^-/-^* null mutants as discussed above.

4) Then a more theoretical point? Have the authors considered that the difference is more in the stability of the abnormal structures. Let's say, without a cartwheel and maybe enough PLK4 activity and high level of other centriolar components, the centrioles are abnormally assembled- they have no cartwheel, but they are disassembled very fast in the embryo but not in ESCs?

We agree and share the reviewer’s interpretation for the potential requirement of SAS-6 *in vivo* to stabilize intermediate structures, that is likely compensated for by the richer centrosome composition (Figure 6) and potent PLK4 activity (Figure 7) in mESCs. The new data support this mechanistic interpretation, which was not directly discussed in the first version of the manuscript and is included in the revised version.

5) Even if there is a real difference and without Sas-6 ESCs can make centrioles that are abnormal in structure and function (at least at the cilia assemble level), the choice of words "strictly required", I am not sure it is correct. Because, since Sas-6 is described by many studies as the factor required for cartwheel assembly, which occurs very early in the pathway, this means that in mESCs centrioles can assembled without forming a cartwheel. And so that the cartwheel is actually not required for the initial building block, but more as a structure that maintains the whole centriole in an intact manner?

We agree with the reviewer on the likely requirement of SAS-6, and therefore the cartwheel as a whole, for the symmetry and integrity of the forming centrioles, which is along the same line as in point #4. In our interpretation, “centriole formation” does not necessarily mean centriole “initiation” but rather the ultimate presence of the centriole as a structure. We used more specific wording to match our shared interpretation with the reviewer.

6) The authors mentioned that in flies, abnormal Sas-6 structures have been described in certain cell types. Are these mutants, null mutants? In other words, do these structures assembled in a context of no Sas6 or abnormal Sas-6 protein or even low levels of Sas-6?

According to the published report (Figure 3—figure supplement 1B in Rodrigues-Martins et al., 2007, PMID: 17689959) the fly brains have no detectable DSAS-6 protein. Therefore, we assume that they are *Sas-6* null fly mutants. The abnormal centrioles in *Sas-6 C. Reinhardtii* mutants and *Sass6^-/-^* mESCs null mutants support the conclusion that the main role of SAS-6, and perhaps the cartwheel, is in maintaining the integrity of the forming procentriole.

Other points:I think the 1st sentence of the abstract appears disconnected from the rest. The same goes for the 1st sentence of the introduction. And also, what is the evidence that pluripotent stem cells rely primarily on the proper assembly of a mitotic spindle? They rely on many other things, not sure this is the first one.

The sentences were removed per the reviewer’s suggestion.

The authors mention that centrioles are lost in Sas6-/- after "differentiation" of mESCs. The term differentiation is not appropriate, and confusing here. Differentiation normally refer to cells that stopped proliferating and exited the cell cycle, which is not the case here, as NPCs are progenitor cells that keep cycling.

We believe the reviewer is referring to “terminal differentiation”, when the cells exit the cell cycle and adopt their destined cell fates. The word “differentiation” in this context refers to limiting the potency of stem cells into a subset of cell fates such as NPCs, which are proliferating progenitors.

Figure S1: Percent of cells with centrosomes was assessed by a co-staining of γtubulin and Cep164, which mark the mother centrioles. As Cep164 may be absent from centrosomes after lack of centriole maturation in sas6-/- embryos, another combination of staining should be performed to evaluate the percent of cells without centrosomes. γtubulin staining can be seen in Sas6 em5/em5 embryos, while the quantification claims total absence of centrosomes. The authors use the CENT2-eGFP transgenic line to count the number of centrioles in Figure 3, they should do the same in Figure S1.

We have followed the reviewer’s recommendation of counting CETN2-eGFP for the assessment of centrioles in controls and *Sass6* mutant embryos (Figure 2E, F).

The γ-tubulin (TUBG) aggregates at the poles of dividing cells are assembled in the absence of centrioles, as shown in *Sass6^em5/em5^* embryo sections (Figure 2C). In addition, we have previously observed these pericentriolar material aggregates in *Cenpj* mutant embryos (Bazzi and Anderson, 2014), which do not contain centrioles in serial transmission electron microscopy. Therefore, we do not refer to them as centrosomes in the absence of centrioles at their core.

Reviewer #1 (Significance (Required)):This study shows with a novel mouse model the requirement of centrioles during mouse development. It will be relevant to centrosome labs, the novel mouse lines will be useful to many labs working on centrioles, cilia and centrosomes.My expertise: centrosome biology

We thank the expert reviewer for the critical comments and suggestions, and the positive evaluation of our manuscript.

Reviewer #2 (Evidence, reproducibility and clarity (Required)):Here, Grzonka and Bazzi present their work on characterizing the requirement of SASS6 in mouse embryo development and in embryonic stem cell (mESC) culture. In mouse, female and male gametes lack centrioles, and early divisions occur without centrioles. de novo formation typically happens at the blastocyst stage (~E3.5). The authors generated two SASS6 knock-out strains, SASS6 em4/em4 (frameshift deletion, reported as a severe hypomorphic allele), and SASS6 em5-em5 (frameshift deletion, reported as a null allele). Mutant embryos arrest development at mid-gestation unless the p53, USP28 and USP28 pathway is perturbed. As expected, centrioles do not form in SASS6 -/- mice. However, the authors report that de novo formation of centrioles is facilitated in mESC culture conditions for SASS6 CRISPR knock-out mESCs and mESCs derived from SASS6 em5/em5 blastocysts. Centrioles are lost upon differentiation of SASS6 CRISPR knock-out mESCs into neural progenitor cells (NPCs).The presented study is relevant for scientists investigating the requirements for centriole formation during embryonic development. Further, it provides insights in possibly different requirements for centriole formation between stages of differentiation, as well as differences in in vivo and in vitro models.

We thank the reviewer for finding our work relevant and insightful into the differential requirements for centriole formation depending on the cell type.

The data represented by Grzonka and Bazzi are robust and support the manuscript and conclusions made. However, the study is predominantly descriptive, and the authors do not test the molecular pathway underlying the de novo formation of centrioles observed in SASS6 -/- mESCs. It is generally believed that de novo formation of centrioles is not possible in SASS6 knock-out cells although work from Wang and Tsou with SASS6 a oligomerization mutant suggests otherwise. A dissection of the specific factors required for the de novo formation of centrioles in the mESC context would provide more insights into de novo centriole assembly in general and would increase the impact of this work. I would support publication of the manuscript if the following points are addressed:

We again thank the reviewer for finding the data robust and support our conclusions and interpretation. We agree with the reviewer that our study opens new questions about how mESCs manage to assemble centrioles in the absence of SAS-6. Together with the phenotypes of the *Sas-6* mutant *D. melanogaster* and *C. Reinhardtii*, and the *SAS-6* oligomerization mutants (but not full *SAS-6* mutants) in human cell lines mentioned by the reviewer and cited in our manuscript, the data open new investigations into the exact requirements of SAS-6 and the cartwheel in centriole biogenesis in the different cellular contexts, their centrosome composition and PLK4 activity.

1. One of the main figures, ideally Figure 1, should be dedicated to the characterization of the newly generated mouse strains. This should also be elaborated in the text further. I would like to see a schematic representation of the genomic modifications. The SASS6 stainings of wt and Sas-6 knock-outs (now Figure 1—figure supplement 1F) should be shown in that context as well as the Figures S2A-C. The authors should discuss why there still appears to be SASS6 protein in the SASS6-em5/em5 Sas-6 stainings visible. Also, the western blot, especially the unspecific bands so close to the SAS-6 protein, should be discussed. Adding qRTPCR results would also be good.

Per the reviewer’s requests, we moved the embryo mutant characterization (Figure 2) and mESCs (Figure 4) to the main figures and elaborated further in the text. The genomic modifications in mice are described in a detailed tabular format in Table 1 in Materials and methods. The immunofluorescence staining in Figure 2C was performed on mouse embryonic sections, which tend to have higher backgrounds than cultured cells; Thus, we attribute the very low percentage of SAS-6 staining in *Sass6^em5/em5^* mutants to higher background, especially given the lack of centrioles in these mutants at all the stages examined.

For Western blots, we used different antibodies against SAS-6 that were either commercially available (Proteintech cat# 21377-1-AP, Sigma-Aldrich cat# HPA028187 and Santa Cruz cat# SC-81431) or non-commercial (kind gift from Renata Basto, Institute Curie). The SAS-6 antibody from the Basto lab gave the most reliable and reproducible results. Using this antibody, and in our own interpretation, we were not able to detect SAS-6 by Western blots in *Sass6* mutant mESCs (including hypomorphic alleles). We concluded that SAS-6 in mESCs and mouse embryos is expressed at low levels. Thus, we decided to use the antibody provided by Renata Basto and achieve higher band resolution, as shown in current Figure 2B and Figure 3C, although it shows two thick non-specific bands flanking the specific band for SAS-6.

For a more definitive knockout in mESCs, we decided to bi-allelically delete the entire *Sass6* ORF DNA from the ATG to the TAA (over 34 Kb of DNA, Figure 3A). According to the central dogma of molecular biology, when there is no DNA, then there should be no mRNA (Figure 3B) and no protein (Figure 3C, Figure 3—figure supplement 1A). In confirmation of this premise, RT-PCR data showed that *Sass6* mRNA is not detectable in these *Sass6^-/-^* null mESCs (Figure 3B). Also, Western Blot analyses (Figure 3C) and immunofluorescence analyses (Figure 3—figure supplement 1A) did not detect SAS-6 in these cells.

In addition, we have used the Basto lab antibody for SAS-6 for Western blots on mouse embryos, which detect low levels of SAS-6 in controls and virtually no specific signal in the mutants (Figure 2B).

2. The authors could elaborate on the topic of mESCs as a special in vitro model for centriole biology akin to the more "primitive" origins of life such as algae.

In the discussion, we have elaborated on the topic of mESCs as a special system for centriole biology to stress the findings that mESCs without SAS-6 can still form centrioles, but also that these cells seem to tolerate centriolar aberrations, such as in *Sass6* mutants, or even the loss of centrioles, as in *Cenpj* mutants, without undergoing apoptosis or cell cycle arrest.

3. Figure 4 should show timeline of embryo development, include embryo stages (E3.5, E9 etc.), group together mESCs with corresponding embryonic developmental stage. The Figure can indicate when mESCs were derived from SASS6 em5/em5 blastocysts, when they were stained and indicate the number/state of centriole formation observed.

We have revised the model in the new Figure 8 to accommodate the suggestions of the reviewer, but at the same time tried to avoid overcrowding and diluting the main findings of the study.

4. The work from Wang and Tsou using SAS-6 oligomerization mutants should be better discussed in the context of the work presented here since centriole assembly was not affected per se but structural defects were observed, like is the case in this study.

We elaborated on this finding from Wang et al. in the discussion.

5. The observation that the ability of forming centrioles de novo in NPCs derived from ESCs is lost is interesting but the mechanisms underpinning this differentiation remain unclear. The authors at a minimum should speculate on these further.

We agree with the reviewer and have performed further experiments suggesting that enhanced centrosome composition and PLK4 activity in mESCs vs NPCs may account for the ability of mESCs to use SAS-6-independent centriole biogenesis pathways. This comment is along the same line as the difference in phenotype between the cells in the developing mouse embryo and mESCs, where the NPCs are more akin to the *in vivo* phenotype, and this is also shown in the current model (Figure 8).

Cross-consultation commentsLooks like we are all pretty much in agreement.Reviewer #2 (Significance (Required)):This is a well executed study with no major flaws that builds on similar studies on knocking out centriole components in mouse and other cell types. Although well-executed the study remains descriptive and lacks a clear mechanistic understanding of why de novo centriole assembly is ineffective in NPCs. As it stands the advances this study provides to the centrosome biogenesis field remain incremental.

We thank the reviewer for the compliments about our work and agree that it opens new questions in the field about the precise roles of SAS-6 and the cartwheel in centriole biogenesis.

Reviewer #3 (Evidence, reproducibility and clarity (Required)):In this publication, Grzonka and Bazzi build upon their recent work describing the role of SAS-like protein function in centriole formation during embryonic development. More specifically, they demonstrate that loss of Sas-6 in vivo and in vitro disrupts centriole formation. To this reviewer's surprise, they found that Sas-6 is required for centriole formation in embryos, yet, stem cells form centrioles with disrupted centriole length and ability to template cilia.

We thank the reviewer for highlighting the novel and surprising aspect of our work, which is that *Sass6* mutant mESCs are still able to form centrioles. We would like to stress that SAS-4, from our previously published work, and SAS-6, in this study, are not part of the same protein family and have different structures and roles in centriole formation. The naming has its origin in “Spindle-ASsembly abnormal/defective” mutant screens performed in *C. elegans*. Although the phenotypes are similar *in vivo*, due the lack of centrioles in both cases, only mutations in *Cenpj*, but not in *Sass6*, lead to the lack of centrioles in mESCs.

Likely, this occurs from the residual proteins that existed prior to CRISPR-mediated knockout.

Due to the nature of the surprising finding that *Sass6* mutant mESCs can still form centrioles, we understand the concerns and suggestions of this reviewer and the other reviewers in this regard.

For a more definitive knockout in mESCs, we decided to bi-allelically delete the entire *Sass6* ORF DNA from the ATG to the TAA (over 34 Kb of DNA, Figure 3A). According to the central dogma of molecular biology, when there is no DNA, then there should be no mRNA (Figure 3B) and no protein (Figure 3C, Figure 3—figure supplement 1A). In confirmation of this premise, RT-PCR data showed that *Sass6* mRNA is not detectable in these *Sass6^-/-^* null mESCs (Figure 3B). Also, Western Blot analyses (Figure 3C) and immunofluorescence analyses (Figure 3—figure supplement 1A) did not detect SAS-6 in these cells.

These *Sass6^-/-^* mESCs started from a single cell and have been passaged more than 30-40 times without losing centrioles. This is how knockouts have been and are produced. If this mutant is still not convincing, then we respectfully ask that the reviewers provide their own suggestion on what will be more convincing. In our humble opinion, this *Sass6^-/-^* mESCs line can be used to test the specificity of the antibodies in mouse cells and not the other way around.

Unsurprisingly, they found that Sas-6 loss in the developing mouse activates the 53BP1-USP28-p53 surveillance pathway leading to cell death and embryonic arrest at mid-gestation, similar to their findings in Cenpj knockouts. What remains to be properly elucidated is the mechanistic differences in the requirement for Sas-6 in stem cells versus the embryo, which may be beyond the scope of a short report.

We have performed additional experiments both *in vivo* and *in vitro* and the new data suggest that the potent activity of the master kinase in centriole duplication, PLK4, is required for SAS-6-independent formation of centrioles in mESCs (Figure 7). Specifically, at the relatively low concentration of 100 nM of the PLK4 inhibitor centrinone B, the centrioles in *Sass6^-/-^* mESCs are lost, but not in WT mESCs (Figure 7). The added data in Figure 6 and Figure 7 provide a potential mechanism of how mESCs are able to form centriole in the complete absence of SAS-6.

As it reads, the manuscript is a compliment to their Sas-4 paper but falls short of novelty and providing large strides in revealing the role of centriolar proteins in developmental processes. Moreover, the advances beyond the requirement for centriole and associated proteins in embryology is missing, therefore enthusiasm is tempered. Below are remaining concerns that must be addressed:

We understand the tempered enthusiasm of the reviewer due to the potentially similar phenotypes between the mouse knockout of two different genes, *Cenpj* and *Sass6*, that have been shown to be required for centriole formation in other contexts. We have performed the *in vivo* experiments and confirmed these expectations. Moreover, we performed the analyses suggested by the reviewer (see below). We also better highlighted our work in mESCs, where we observed the surprising differences in the mutant phenotypes between *Cenpj*, which lose centrioles, and *Sass6*, which are still able to form centrioles, albeit abnormal ones.

Remaining concerns:The authors should provide clear description of the embryonic region (neural plate & mesenchym) used to analyze centriole presence or loss in Figures 1 and S1. Was this in the forelimb vs hindlimb regions?

The assessment of centrosomes in Figure 2 and Figure 2—figure supplement 1 was performed on sections from the brachial region, forelimb and heart level, as indicated in the revised Materials and methods.

Similar to their Cenpj-mouse data, the authors should provide data detailing the mitotic index and activation of the mitotic surveillance pathway beyond just p53 staining. As novelty is not the only criteria for publication, a thorough analysis of the Sas-6 activation of the mitotic purveyance pathway should be provided, including the crosses between Sas-6 and p53, 53bp1 and usp28 knockout crosses to demonstrate the pathway functions similarly to Cenpj loss.

We performed the additional experiments requested by the reviewer that are similar to our previous work in *Cenpj* mutants (Xiao*, Grzonka* et al., 2021). We performed these analyses knowing that both *Cenpj* and *Sass6* mutants lose centrioles and activate the mitotic surveillance pathway, as the reviewer indicated. In particular, we quantified the mitotic index in the *Sass6^em4/em4^* and *Sass6^em5/em5^* mutants (Figure 1—figure supplement 1) and performed p53 and Cl. CASP3 staining in the double mutants with *Trp53bp1* or *Usp28*, to confirm that the pathway has been suppressed in these mutants (Figure 1).

Centriole structure should be assessed in the embryos using EM to assess loss and confirm the structural defects. This would strengthen their argument and be a slight advance to their largely descriptive paper.

Because the *Sass6^em5/em5^* embryos lack centrioles, as indicated by regular immunofluorescence and Ultrastructure-Expansion Microscopy (U-ExM), using EM would be an attempt to find a structure that does not exist. In our opinion, it would again be a repetition of TEM studies that we have already performed in *Cenpj^-/-^* mutant embryos, that lack centrioles (Bazzi and Anderson, 2014). Using U-ExM has advanced the centriole biology field to a level that is approaching EM resolution and, in our opinion, can substitute for EM.

The WB for Sas-6 knockout is not convincing and should be redone. There are validated Sas-6 antibodies available from SCBT and Proteintech. It is not clear that the band is gone or if there's overlap with the non-specific band.

The answer to this comment is shown above. In addition, we have used the Basto lab antibody for SAS-6 for Western blots on mouse embryos, which detect low levels of SAS-6 in controls and no signal in the mutants (Figure 2B). We also repeated the SAS-6 Western blots on mESCs for better band resolution as recommended by the reviewer (Figure 3C).

The authors describe the centriolar structural defect in the mESCs in Figure 2C and D, and further characterize the phenotype in S2D-H. Given the role of the SAS6-CEP135-CPAP axis for centriole elongation, it is peculiar that they see elongation upon reduction of CEP135. The authors should find a rationale mechanism to explain their discordant findings. In addition, other centriole distal end components including CEP97 and CP110 should be examined to determine the structural end caping defect in the Sas-6 mESC.

More than 70% of the centrioles in *Sass6^-/-^* mESCs retain CEP135 (Figure 4C, D), but the majority of CEP135 signals (> 60%) seem to be abnormally localized (Figure 4—figure supplement 1B, C). One potential explanation for the elongated centrioles in *Sass6^-/-^* mESCs is that the mis-localization of CEP135 impacts on the integrity of the centriole and results in parts of the centriolar walls being elongated. Per the reviewer’s suggestion, we have performed U-ExM with immunostainings for CP110 (Figure 4F, G) or CEP97 (Figure 4—figure supplement 1D, E), that also regulate centriole capping and elongation. The data indicated that similar to WT mESCs, they mostly localize to the ends of the abnormal centrioles in *Sass6^-/-^* mESCs.

In Figure 2I, J the authors state the ciliation rate for the WT mESCs was only 11%, could the authors provide an explanation for the low ciliation rate in WT mESCs? Could cells be arrested to increase the ciliation rate? In addition, is there a rational explanation for the loss of centrioles and centrosomes upon differentiation into NPCs?

mESCs ciliation rate has been shown to be generally low (Bangs et al., 2015; Xiao et al., 2021) perhaps because the cells spend most of the cell cycle in the S-phase. mESCs require a high serum percentage and well-defined media for growth and maintenance. In our hands, attempting to arrest the cells by withdrawing serum, or reducing its percentage, resulted in cell death and a change in morphology to the differentiated phenotype (unpublished data). Our data indicate that a pluripotent state in *Sass6^-/-^* mESCs is compatible with centriole formation but differentiation to NPCs results in the loss of centrioles. Therefore, we have refrained from interfering with the cell cycle of mESCs in order to avoid these confounding effects on cellular viability and centriole formation.

Regarding the loss of centrioles upon differentiation of *Sass6^-/-^* mESCs into NPCs, we agree with the reviewer and have performed additional experiments that showed that centrosomal proteins (TUBG, CEP152, SAS-4 and STIL) are more abundant in the interphase centrosomes of mESCs compared to NPCs (Figure 6). Moreover, our new data suggest that the potent activity of the master kinase in centriole duplication, PLK4, is required for SAS-6-independent formation of centrioles in mESCs (Figure 7). Specifically, at the relatively low concentration of 100 nM of the PLK4 inhibitor centrinone B, the centrioles in *Sass6^-/-^* mESCs are lost, but not in WT mESCs (Figure 7). Our new data in Figure 6 and Figure 7 provide a potential mechanism of how mESCs are able to form centriole in the complete absence of SAS-6.

In figure 3F in the Sas-6−/− NPCs have a box around a cell without centrosomes yet in 3G here are some cells with centrosomes. While the authors are trying to demonstrate the decrease in centrosome in the Sas-6−/− NPCs, they should show the few cell that have centrosomes or centrosome-like structures.

We added an example for the minority of cells (less than 10%) that retain centrosomes upon differentiation of *Sass6^-/-^* mESCs into NPCs (Figure 5—figure supplement 1B).

Cross-consultation commentsAs mentioned in my review; while the Sas6 model is new, it does not provide further evidence of why centriole duplication is important in developing mice aside from it causing an abortive mitosis leading to cell death. The discordant phenotype in the mESCs likely arises from residual Sas6, similar to experiments that were performed in flies with Sas-4 depletion. Moreover, the odd centriole phenotype represents a very small number of cells and is likely phenomenological. In addition, their work from last year demonstrated a clear connection between Cenpj loss leading to the mitotic surveillance pathway activation. They performed double knockouts that partially rescued the survival phenotype. This new work falls short of that publication.

Reviewer #3 (Significance (Required)):The new publication adds a known component to the list of animal models for centrosome-opathies but fails to provide novel mechanistic insights. Dr. Bazzi's publication on Sas-4 was far more novel at the time of publication due to the multiple mouse crosses that could rescue the phenotypes. The recent publication fails to provide as much evidence or any novel insights into the role of Sas-6 (sufficient to be convincing).The audience will be limited to centrosome biologists and even then it may not have enough novelty to be compelling. I would recommend with the revisions to be published in a more specialized journal.My expertise lies in genetic causes of microcephaly-associated with mutations in centrosome encoding proteins.

We thank the reviewer for taking the time to evaluate our work and provide helpful comments and suggestions. We would like to emphasize that even if a certain phenotype is expected, the experiment has to be performed to test the hypothesis, which is the case with the *Sass6* mutant embryos phenocopying the *Cenpj* mutants. In our opinion, the novelty of our work goes beyond the *in vivo* knockout and analyses of *Sass6* mouse embryo mutants to the ability of *Sass6^-/-^* null mESCs to form centrioles. This surprising finding opens new avenues of investigation into the precise roles of SAS-6, the cartwheel, centrosome composition and thresholds of PLK4 activity in centriole biogenesis. We are confident that our study will provide a compelling evidence to re-examine these roles in other cell types and organisms.

Description of analyses that authors prefer not to carry out

We performed almost all of the analyses requested by the reviewers and added more mechanistic data to our study as elaborated above.